# Interaction effects of the 5-HTT and MAOA-uVNTR gene variants on pre-attentive EEG activity in response to threatening voices

Róger Marcelo Martínez [1,2,10], Tsai-Tsen Liao[3,4,10], Yang-Teng Fan[5], Yu-Chun Chen[6] & Chenyi Chen [1,7,8,9]✉

Both the serotonin transporter polymorphism (*5-HTTLPR*) and the monoamine oxidase A gene (*MAOA-uVNTR*) are considered genetic contributors for anxiety-related symptomatology and aggressive behavior. Nevertheless, an interaction between these genes and the pre-attentive processing of threatening voices –a biological marker for anxiety-related conditions– has not been assessed yet. Among the entire sample of participants in the study with valid genotyping and electroencephalographic (EEG) data (N = 140), here we show that men with low-activity *MAOA-uVNTR*, and who were not homozygous for the *5-HTTLPR* short allele (s) (n = 11), had significantly larger fearful MMN amplitudes –as driven by significant larger ERPs to fearful stimuli– than men with high-activity *MAOA-uVNTR* variants (n = 20). This is in contrast with previous studies, where significantly reduced fearful MMN amplitudes, driven by increased ERPs to neutral stimuli, were observed in those homozygous for the *5-HTT* s-allele. In conclusion, using genetic, neurophysiological, and behavioral measurements, this study illustrates how the intricate interaction between the *5-HTT* and the *MAOA-uVNTR* variants have an impact on threat processing, and social cognition, in male individuals (n = 62).

[1] Graduate Institute of Injury Prevention and Control, College of Public Health, Taipei Medical University, Taipei, Taiwan. [2] School of Psychological Sciences, National Autonomous University of Honduras, Tegucigalpa, Honduras. [3] Graduate Institute of Medical Sciences, College of Medicine, Taipei Medical University, Taipei, Taiwan. [4] Cell Physiology and Molecular Image Research Center, Wan Fang Hospital, Taipei Medical University, Taipei, Taiwan. [5] Graduate Institute of Medicine, Yuan Ze University, Taoyuan, Taiwan. [6] Department of Physical Education, National Taiwan University of Sport, Taichung, Taiwan. [7] Research Center of Brain and Consciousness, Shuang-Ho Hospital, Taipei Medical University, New Taipei City, Taiwan. [8] Graduate Institute of Mind, Brain and Consciousness, College of Humanities and Social Sciences, Taipei, Taiwan. [9] Psychiatric Research Center, Wan Fang Hospital, Taipei Medical University, Taipei, Taiwan. [10] These authors contributed equally: Róger Marcelo Martínez, Tsai-Tsen Liao. ✉email: chenyic@tmu.edu.tw

The serotonin transporter gene (*SLC6A4*) possesses a functional polymorphism (*5-HTT*) in its linked polymorphic region (*5-HTTLPR*), and which has been observed to be a genetic contributor towards anxiety-related traits and/or symptomatology[1]. This is due to such polymorphic region being able to contain two variants—one short (S allele) and one long (L allele)–, and which affects differently the way in which the serotonin transporter behaves. The S allele encodes less *5-HTT* mRNA and protein in terms of quantity, which leads to the transporter carrying significantly less serotonin—relative to the L allele— back to the presynaptic neuron from the synaptic cleft; thus, the remaining excess of serotonin prolonging serotonergic receptor excitation[2].

Consequently, several neuroscientific studies delving into the research of the differential brain activations in S -and L-allele carriers, have observed that those carriers of the S allele incur in amygdala hyperactivity[3]. This overexcitation not only has been associated to a susceptibility towards neuroticism or negative emotionality[4–6] but, furthermore, is the driving factor behind a mechanism encompassing an amygdala-related heightened baseline level of arousal even to nonthreatening stimuli, and whose outcome appears to be anxiogenic symptomatology[7].

Alongside, the monoamine oxidase A gene (*MAOA-uVNTR*) has been observed to be paramount in the catabolism of neuroactive amines such as serotonin, norepinephrine and dopamine. In studies conducted with a line of transgenic mice where their *MAOA-uVNTR* encoding gene was deleted, mice pups exhibited alterations in behavior, exhibiting trembling and fearfulness; while adult mice exhibited markedly different behavioral alterations, characterized by enhanced aggression in males. This was due to the pups' brain serotonin levels increasing up to nine times the average level, while the pups as well as the adults' norepinephrine concentrations increasing up to twice the average levels[8].

In humans, Brunner, et al.[9] study was the first to establish an association between a genetic abnormality in the *MAOA-uVNTR* gene and a repeated incidence across generations of a Dutch family regarding criminal violent behavior perpetrated by males. Nevertheless, Brunner[10] himself cautioned on baptizing the *MAOA-uVNTR* gene as an "aggression gene", since this gene's influence was effected through a highly complex relationship to other neurotransmitter function, rather than by the gene itself (and as attested by the aforementioned mice studies). In the same line, more recent studies probing into the *MAOA-uVNTR* gene's allelic variants have found that those variants generating low (*MAOA-L*) enzymatic activity have a moderating effect between the association of early adverse life experiences (such as child maltreatment, early trauma, material deprivation, school failure, among others) and aggression[11–14], as well as aggression under specific circumstances (such as aggression following provocation)[15], and when compared to the variants generating high enzymatic activity (*MAOA-H*). Furthermore, recent neuroimaging studies demonstrated that *MAOA-L* males exhibit increased amygdala reactivity, and decreased prefrontal activity during emotional arousal[16,17].

Given that human voices transmit invaluable social information[18], this study utilized Mismatch negativity (MMN)—which is an auditory event-related potential (ERP) elicited by an odd (deviant) stimulus embedded in a series of repetitive stimuli (standard)– as evoked through a passive auditory oddball paradigm. Participants were asked to engage in a task, while task-irrelevant sound stimuli were presented to them in the background and in a quasi-random fashion, and with the standard stimuli occurring with more frequency than the deviant[19]. MMN has previously been used to successfully demonstrate a positive association between MMN amplitudes and anxiety-like symptomatology, as it reflects the emotional hypervigilance which characterizes anxiety[20,21]. As such,

MMN is able to index the neurobiological processes sitting at the border between attention-dependent, and automatic mechanisms, which regulate the access to higher orders of memory and conscious perception[22]. Due to this ability to draw unto memory and attentional processes, it has been postulated that emotional MMN (eMMN)—an MMN subtype that uses emotionally spoken syllables as stimuli in the auditory oddball paradigm[23]—can successfully assess the automatic neural processing of emotional voices in as early as the pre-attentive stage[24,25]. Furthermore, a processing chain encompassing the primary auditory pathway, neural structures involved in cognition and emotion—e.g., the orbitofrontal cortex, amygdala, superior temporal gyrus and sulcus—and in addition to the saliency network (insula), has been uncovered[25–28]. Particularly, research eliciting MMN through the use of threatening syllables as stimuli, was shown to significantly elicit amygdala activity[27]. What's more, MMN elicited by means of fearful voices has been observed as being able to predict anxiety-related symptomatology[25]. These findings provide further support for the assumption that eMMN can prod voice processing per se, disentangling attentional modulation from emotional salience. We thus assume that eMMN can very well reflect *5-HTT*- and *MAOA-uVNTR* -dependent neural modulation.

While the *MAOA-L* gene was found to be associated with a hyper-responsiveness to threatening stimuli and fearless temperament in men[17,29], this overexcitation to threatening emotions was also recognized as a pivotal bio-maker for *5-HTTLPR*-related anxiogenic symptomatology[7] –condition particularly prevalent among the women population alongside depression and somatic complaints[30]. In order to examine the interaction effect of the two serotonin modulating genes, the *5-HTTLPR* and *MAOA-uVNTR*, on the perception of threatening stimuli, and to test whether this genetic interaction co-varied with the factor gender, this study genotyped the *5-HTTLPR* and *MAOA-uVNTR*, and recorded the MMN elicited by emotionally spoken syllables in healthy male and female volunteers with varying degrees of state and trait anxiety. Meanwhile, while fear and anger are the two most commonly studied threat-related negative emotions that have been identified in previous studies as being able to elicit amygdala activity even without conscious awareness[31], these two emotions vary as a function of adaptability in terms of the human defense system. While anger is an approach-motivated negative emotion, fear acts as an avoidance-motivated negative emotion. We further examined the above-mentioned gender-gene interaction in electroencephalography (EEG) response to angry and fearful voices. Through the incorporation of multimodal indices—including genetic, neurophysiological, and behavioral measurements—this study explores the possibility of a gene × gene × environment interaction as the starting point leading towards variation in social behavior.

Based on our previous findings regarding the amygdala-related heightened baseline level of arousal even to nonthreatening stimuli as the neural mechanism underlying anxiety[7], we hypothesized that *MAOA-L* and S allele carrying subjects would both exhibit distinct eMMN when compared to *MAOA-H* and non-carrier subjects, and that the formers' eMMN would be equally associated with anxiety-related symptomatology. We further hypothesized that *MAOA*-L subjects will exhibit—as S carriers did—weaker eMMN amplitudes as a function of increased ERPs elicited by baseline neutral stimuli.

## Results

**Genotyping distribution and STAI**. The 5-*HTTLPR* was found to have allele frequencies of S, $n = 199$ (71.1%); LA, $n = 29$ (10%); and LG, $n = 52$ (18.6%), and a genotype distribution of S/S, $n = 78$ (55.7%); S/LG, $n = 30$ (21.4%); LG/LG, $n = 6$ (4.3%);

S/LA, $n = 13$ (9.3%); LG/LA, $n = 10$ (7.1%); and LA/LA, $n = 3$ (2.1%). Genotype distribution of the 5-HTTLPR of this sample was deviated from Hardy–Weinberg equilibrium, $\chi^2(3) = 10.602$, $P = 0.014$ partially due to the extremely low cases of LA/LA and the significantly higher S/S to L/L ratio which was previously identified in Han Chinese, compared to that observed in western Caucasian populations[7,32]. However, they had no significant deviation from the sample in wave 1 data collection, where the data was collected between the two calendar dates of 10/11/2016 and 07/02/2017, and was later published in March 2020 [$\chi^2(5) = 0.09$, $P = 0.99$][7]. The following analyses employed the genotype groups: SS = 78, and LL/LS = 62. The self-evaluation of anxiety from the STAI ranged from 23 to 72 (mean ± SD: 42.81 ± 10.29) in the trait anxiety and from 20 to 58 (35.21 ± 8.65) in the state anxiety (Table 1). The 5-HTTLPR genotype was not different across age ($P = 0.29$), gender (male % of total: 40% vs. 50%; $P = 0.23$), STAI-T ($P = 0.24$), and STAI-S ($P = 0.34$).

For the genetic variation of MAOA-uVNTR in this Taiwanese sample, we identified MAOA-uVNTR genotypes and allelic frequencies by following a new classification method for the sequence repeats, namely, 2.5 R, 3.5 R, 4.5 R, and 5.5 R[33–35], which corresponds to 2 R, 3 R, 4 R, and 5 R in previous studies with the original method[36–38]. The difference lies in the first 15 bp half-repeat sequence ($-1141/-1127$ bp), which is next to the repeated 30 bp sequence ($-1262/-1142$ bp)[39]; the redefined classification includes this sequence (Fig. 1a). Eight different MAOA-uVNTR genotypes were detected in our population, including two genotypes in men and six genotypes in women (Table 1). The polymerase chain reaction (PCR) products of the observed types of MAOA-uVNTRs are shown in Fig. 1b. Among the four MAOA-uVNTR alleles detected, the 3.5 R allele was the most prevalent (Table 2). The 2.5 R and 5.5 R allele appeared only in heterozygous genotypes, such as 2.5/3.5, 3.5/5.5, and 4.5/5.5 in women, and at a very low frequency, parallel to a recent Korean study[34]. Of note, while most MAOA-uVNTR polymorphism studies that were based on western Caucasian population identified an reversed-U shape quadratic association between the number of repeats and the promotor activity of MAOA-uVNTR alleles where 3.5 R and 4 R variants showed significantly higher activity than 3 R or 5 R[36,38], studies employed Taiwanese and Koreans samples revealed an opposite pattern[34,40]. 4 R/4.5 R variants showed significantly lower transcriptional activity (analyzed by the luciferase-reporter assay) than 3 R/3.5 R or 2 R/2/5 R in East Asian population. The following analyses employed the MAOA-uVNTR genotype groups based on the findings of Asian data.

Since MAOA-uVNTR is an X-linked gene, men can only be classified by having high or low activity, but women can be classified as having high (H), intermediate (M) or low (L) MAOA-uVNTR activity. The genotype frequencies for males were L: 38.7% and H: 61.3%; for females they were L: 12.8%, M: 50%, H: 37.2%. The genotype distribution of the MAOA-uVNTR was in Hardy–Weinberg equilibrium [$\chi^2(6) = 1.904$, $P = 0.928$]. The MAOA-uVNTR genotype was not different across age (male: $P = 0.13$; female: $P = 0.91$), STAI-T (male: $P = 0.77$; female: $P = 0.69$), and STAI-S (male: $P = 0.95$; female: $P = 0.81$) in both male and female participants.

**Neurophysiological measures of preattentive discrimination of fearful and angry voices.** Emotional MMN (eMMN) was determined by subtracting the neutral ERPs from angry and fearful ERPs (Fig. 2a). The four-way mixed ANOVA, comprising gender (male or female) as the between-subjects factor, and the deviant type (fearful or angry), coronal site (left, midline, right) and anterior-posterior site (frontal or central) as the within-subjects factors, revealed a main effect of coronal site (left, midline, right) [$F_{2,276} = 7.09$, $P = .001$, $\eta p^2 = 0.049$, $(1-\beta) \approx 100\%$] as well as a marked trend of deviant type (fearful vs. angry) [$F_{1,138} = 3.53$, $P = 0.062$, $\eta p^2 = 0.025$ $(1-\beta) = 96.43\%$]. While midline (2.45 ± 0.22 μV, $P < 0.001$) and right site electrodes (2.05 ± 0.21 μV) showed significant larger MMN amplitudes than the left electrodes (2.35 ± 0.22 μV), fearful MMN (2.47 ± 0.24 μV) showed higher amplitudes than angry MMN (2.09 ± 0.21 μV) that appeared to be marginally significant ($P = 0.062$).

Significant interactions were observed among the deviant type and anterior–posterior site [$F_{1,138} = 5.21$, $P = 0.024$, $\eta p^2 = 0.036$, $(1-\beta) = 99.5\%$], and among the deviant type, anterior–posterior site, and gender [$F_{1,138} = 3.98$, $P = 0.048$, $\eta p^2 = 0.028$, $(1-\beta) = 97.87\%$]. Post hoc analyses revealed that the fearful MMN (2.66 ± 0.27 μV, $P = 0.015$) exhibited significant higher amplitudes than angry MMN (2.09 ± 0.26 μV) only in frontal but not in central electrodes (fearful MMN:

**Table 1 Demographic and descriptive statistics of 5-HTT genotypes in the Taiwanese population (mean ± SD).**

| 5-HTT genotypes | | | Age | STAI-S | STAI-T |
|---|---|---|---|---|---|
| | | n (%) | | | |
| Men | LL | 11 (18) | 23.1 ± 4.1 | 35.5 ± 5 | 41.4 ± 8.3 |
| | LS | 20 (32) | 23.8 ± 2.4 | 32.7 ± 9.1 | 39.6 ± 12.7 |
| | SS | 31 (50) | 24.3 ± 5.1 | 34.8 ± 8.9 | 42.9 ± 10.1 |
| | Sub-total | 62 (100) | 23.9 ± 4.2 | 34.2 ± 8.4 | 41.5 ± 10.7 |
| Women | LL | 8 (10) | 23.2 ± 1.7 | 36.1 ± 5.5 | 45.9 ± 8 |
| | LS | 23 (29) | 25.5 ± 6 | 34.9 ± 9 | 42.2 ± 9.1 |
| | SS | 47 (60) | 23 ± 2.5 | 36.5 ± 9.3 | 44.3 ± 10.7 |
| | Sub-total | 78 (100) | 23.7 ± 4 | 36 ± 8.8 | 43.8 ± 9.9 |
| Total (n/%) | | 140 (100) | 23.8 ± 4.1 | 35.2 ± 8.6 | 42.8 ± 10.3 |

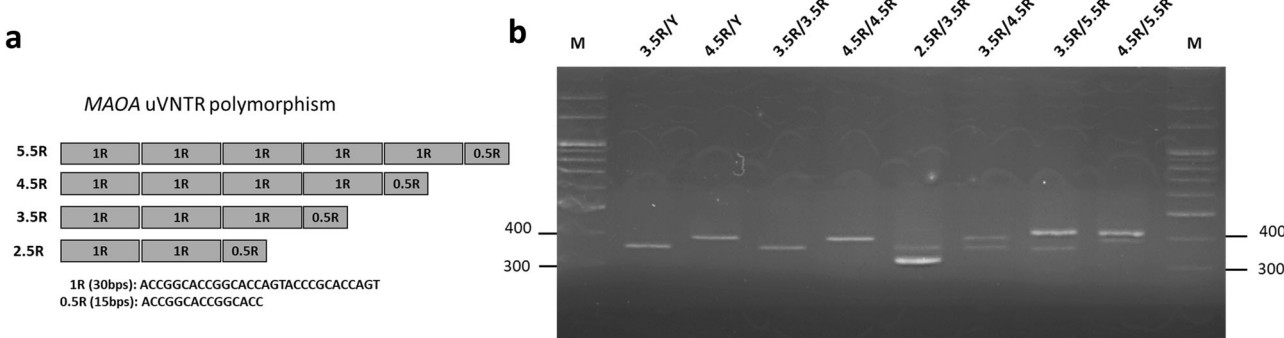

**Fig. 1 Genotyping of the MAOA-uVNTR polymorphism. a** This diagram shows MAOA-uVNTR alleles 2.5 R, 3.5 R,4.5, and 5.5 R. 1 R of each MAOA-uVNTR allele consisted of a 30 bp, and the 0.5 R consisted of a 15 bp. **b** The representative image of MAOA-uVNTR genotypes in 3% agarose gel. Lanes 1 and 2 are 3.5 R/Y and 4.5 R/Y from men; lanes 3, 4, 5, 6, 7 and 8 are 3.5 R/3.5 R, 4.5 R/4.5 R, 2.5 R/3.5 R, 3.5 R/4.5 R, 3.5 R/5.5 R, and 4.5 R/5.5 R from women, respectively. Electropherogram of alleles showing 2.5, 3.5, 4.5 and 5.5 repeats.

**Table 2 Demographic and descriptive statistics of MAOA-uVNTR genotypes in the Taiwanese population (mean ± SD).**

| MOAO genotypes | | n (%) | Alleles | | | | Age | STAI-S | STAI-T |
|---|---|---|---|---|---|---|---|---|---|
| | | | 2.5 R | 3.5 R | 4.5 R | 5.5 R | | | |
| Men | *Hemizygous* | | | | | | | | |
| | 3.5 R/Y | 38 (61) | 0 | 38 | 0 | 0 | 23.3 ± 2.6 | 34.2 ± 8.3 | 41.9 ± 10.3 |
| | 4.5 R/Y | 24 (39) | 0 | 0 | 24 | 0 | 24.9 ± 5.9 | 34.3 ± 8.6 | 41.9 ± 11.4 |
| | Sub-total | 62 (100) | 0 | 38 | 24 | 0 | 23.9 ± 4.2 | 34.2 ± 8.4 | 41.5 ± 10.7 |
| Women | *Homozygous* | | | | | | | | |
| | 3.5R/3.5R | 26 (33) | 0 | 52 | 0 | 0 | 23.1 ± 3.2 | 37 ± 8.4 | 44.4 ± 7.8 |
| | 4.5 R/4.5 R | 10 (13) | 0 | 0 | 20 | 0 | 23.8 ± 4.5 | 35 ± 11 | 41.4 ± 9.4 |
| | *Heterozygous* | | | | | | | | |
| | 2.5 R/3.5 R | 1 (1) | 1 | 1 | 0 | 0 | 26 | 39 | 51 |
| | 3.5 R/4.5 R | 39 (50) | 0 | 39 | 39 | 0 | 24.1 ± 4.4 | 35.3 ± 8.9 | 43.4 ± 11.3 |
| | 3.5 R/5.5 R | 1 (1) | 0 | 1 | 0 | 1 | 24 | 31 | 45 |
| | 4.5 R/5.5 R | 1 (1) | 0 | 0 | 1 | 1 | 22 | 46 | 60 |
| | Sub-total | 78 (100) | 1 | 93 | 60 | 2 | 23.7 ± 4 | 36 ± 8.8 | 43.8 ± 9.9 |
| Total (n/%) | | 140 (100) | 1 | 131 | 84 | 2 | 23.8 ± 4.1 | 35.2 ± 8.6 | 42.8 ± 10.3 |

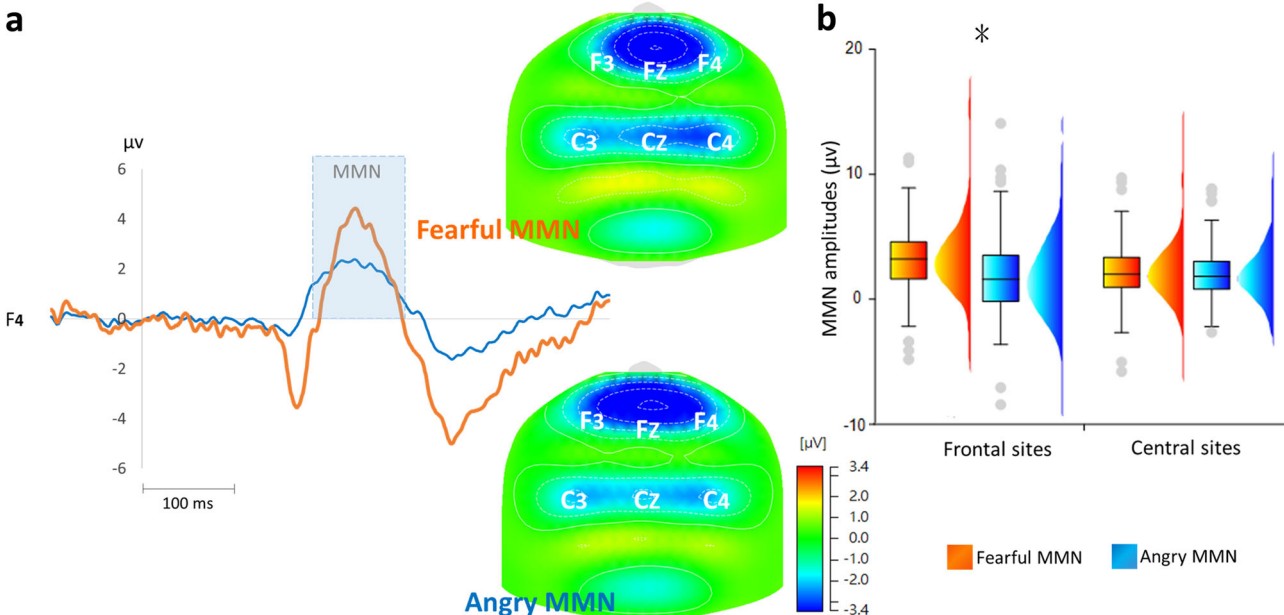

**Fig. 2 Fearful and angry MMN. a** Fearful and angry MMN were derived by subtracting neutral ERPs from fearful and angry ERPs, respectively. **b** eMMN was statistically identified by using a mixed ANOVA comprising gender (male or female) as the between-subjects factor, and the deviant type (fearful or angry), coronal site (left, midline, right) and anterior–posterior site (frontal or central) as the within-subjects factors. Fearful MMN (2.66 ± 0.27 μV, $P = 0.015$) exhibited significant higher amplitudes than angry MMN (2.09 ± 0.26 μV) in frontal electrodes but not in central electrodes (fearful MMN: 2.29 ± 0.23 μV; angry MMN: 2.15 ± 0.2 μV, $P = 0.475$). Whisker boundaries were set as box edge ± 1.5 interquartile range (IQR). Source data are presented in Supplementary Data 1.

2.29 ± 0.23 μV; angry MMN: 2.15 ± 0.2 μV, $P = 0.475$) (Fig. 2b, Supplementary Data 1). The significant interaction between the deviant type and anterior–posterior site was found exclusively in female [$F_{1,77} = 8.23$, $P = 0.005$, $\eta p^2 = 0.097$, $(1-\beta) \approx 100\%$] but not in male participants [$F_{1,61} = 0.054$, $P = 0.817$, $\eta p^2 = 0.001$, $(1-\beta) = 7.8\%$].

**The effects of MAOA-uVNTR and 5-HTT genotypes on eMMN.** In order to examine whether *5-HTT* gene variants covariate with the functional role of the *MAOA-uVNTR* alleles, with both encoding proteins for central serotonergic functions, we further examined the interaction effect between the *5-HTT* and the *MAOA-uVNTR* genotype on the EEG activities in response to

the preattentive processing of threatening voices. Specifically, the mean amplitudes of fearful MMN and angry MMN from midline and right site electrodes (FZ/CZ, F4/C4)—where the largest ERPs were observed—were extracted and subjected into a two-way ANOVA. The *MAOA-uVNTR* genotypes (High vs. Low) and *5-HTT* genotype (SS vs. LL/LS) were the between-subject factors. While the *MAOA* is an X- linked gene, results were presented for men and women.

*Fearful MMN.* Due to the effect of the *MAOA-intermediate* in female participants remaining elusive, we examined the *MAOA-uVNTR* genotypes in women in an explorative manner. Specifically, we tested the *MAOA* effect by treating it as a three-level variable (*MAOA-high* vs. *MAOA-intermediate* vs. *MAOA-low*), a two-level

variable by regrouping *MAOA-high* and *MAOA-intermediate* (*MAOA-high/intermediate* vs. *MAOA-low*), or a two-level variable by regrouping *MAOA-intermediate* and *MAOA-low* (*MAOA-high* vs. *MAOA-intermediate/low*). Because the pattern of results was similar across the above-mentioned three models, we presented the results here for the *MAOA-high/intermediate* vs. *MAOA-low* model of our female participants, with the purpose of group size equalization. The interaction effect of *MAOA-uVNTR × 5-HTT* was significant in male [$F_{1,58} = 6.26$, $P = 0.015$, $\eta p^2 = 0.097$, $(1-\beta) = 71.83\%$] but not in female participants [$F_{1,74} = 0.399$, $P = 0.529$, $\eta p^2 = 0.005$, $(1-\beta) = 9.48\%$].

In men, MOAO exerted its effect only in ones who don't possess a *5-HTT* genotype of SS [$T(29) = 2.33$, $P = 0.027$]. Men with *MAOA-L* ($3.74 \pm 0.97$ μV) showed significant larger fearful MMN than men with *MAOA-H* ($1.84 \pm 0.3$ μV) in the *5HTT*-LL/LS group, whereas they are parallel in the *5HTT*-SS group [*MAOA*-L: $1.66 \pm 0.49$ μV; *MAOA*-H: $2.88 \pm 0.72$ μV; $T(29) = -1.3$, $P = 0.2$] (Fig. 3a, Supplementary Data 2). Furthermore, since fearful MMN stands for the differential amplitudes (ΔERPs) between fearful deviant ERPs and neutral standard ERPs, previous study has revealed that both fearful and neutral ERP responses independently contributed to the fearful MMN[7]. In order to further examine whether this enlarged fearful MMN, that was found in men with *MAOA*-L and *5-HTT*-LL/LS, was due to the increased responses to fearful syllables, reduced neutral ERPs, or both, we conducted one-way ANOVAs comprising *MAOA-uVNTR* genotype (high vs. low) as the between-subjects factor at the electrodes that showed the largest fearful MMN for this particular group of participants. The *MAOA*-low-modulated fearful MMN was associated with the increased fearful ERPs [FZ: *MAOA*-L vs. *MAOA*-H: $2.58 \pm 1.03$ μV vs. $0.96 \pm 0.38$ μV; $T(29) = 1.77$, $P(one-tailed) = 0.044$; C4: $2.72 \pm 1.07$ μV vs. $1.12 \pm 0.3$ μV; $T(29) = 1.82$, $P(one-tailed) = 0.039$] but not with the neutral ERPs (all $P > 0.2$).

Correlation analyses, conducted against ERPs and fearful MMN amplitudes, were used to examine whether the EEG activity to neutral standards or fearful deviants was modulated by *MAOA-uVNTR* gene, independent of gender and *5-HTT* genotype. In participants with *MAOA*-L (combined men and women, $n = 34$), fearful MMN was only positively correlated with the amplitudes of fearful ERP ($R_{34} = 0.82$, $P < 0.001$) but not correlated with neutral ERP ($R_{34} = -0.18$, $P = 0.31$). However, in the rest of participants who have *MAOA*-H or *MAOA*-I (combined men and women, $n = 106$), fearful MMN was positively correlated with the amplitudes of fearful ERP ($R_{106} = 0.64$, $P < 0.001$) and negatively correlated with neutral ERP ($R_{106} = 0.26$, $P = 0.007$) (Fig. 4a, Supplementary Data 3).

*Angry MMN*. As for the angry MMN, there was no main effect of *MAOA-uVNTR* [male: $F_{1,58} = 0.042$, $P = 0.838$, $\eta p^2 = 0.001$, $(1-\beta) = 5.69\%$; female: $F_{1,74} = 0.025$, $P = 0.874$, $\eta p^2 < 0.001$, $(1-\beta) < 5\%$], *5-HTT* [male: $F_{1,58} = 0.84$, $P = 0.363$, $\eta p^2 = 0.004$, $(1-\beta) = 7.81\%$; female: $F_{1,74} = 3.901$, $P = 0.052$, $\eta p^2 = 0.05$, $(1-\beta) = 51.62\%$], nor *MAOA-uVNTR × 5-HTT* interaction [male: $F_{1,58} = 3.247$, $P = 0.077$, $\eta p^2 = 0.053$ $(1-\beta) = 44.95\%$; female: $F_{1,74} = 0.288$, $P = 0.593$, $\eta p^2 = 0.004$, $(1-\beta) = 8.57\%$]. While the main effect of *5-HTT* in women showed a marginal trend toward significance ($P = 0.052$), with a medium effect size, the planned pairwise comparison indicated that women with *5-HTT*-SS tended to have a higher angry MMN amplitude than women with *5-HTT*-LL/LS ($3.084 \pm 0.424$ μV, $1.788 \pm 0.501$ μV, $P = 0.052$, respectively).

Despite the interaction between the *MAOA-uVNTR* and *5-HTT* genes showing a marginal trend toward significance in men ($P = 0.077$), the planned pairwise comparisons did not reveal any significant simple main effect (all $P > 0.1$); albeit the MOAO effect in male participants dependent on the *5-HTT* genotype showing that men with *MAOA*-L yielded increased angry MMN among those men who do not possess a *5-HTT* genotype of SS (*MAOA*-

L: $2.72 \pm 0.95$ μV; *MAOA*-H: $1.76 \pm 0.22$, $P = 0.34$), but decreased angry MMN in those men with the *5HTT*-SS (*MAOA*-L: $1.08 \pm 0.53$ μV; *MAOA*-H: $2.29 \pm 0.7$ μV, $P = 0.18$) (Fig. 3b).

Correlation analyses, conducted against ERPs and angry MMN amplitudes, showed that in participants with *MAOA*-L (combined men and women, $n = 34$), angry MMN was positively correlated with the amplitudes of angry ERP ($R_{34} = 0.71$, $P < 0.001$) and negatively correlated with neutral ERPs ($R_{34} = -0.37$, $P = 0.03$). However, in the rest of participants who have *MAOA*-H or *MAOA*-I (combined men and women, $n = 106$), angry MMN was only positively correlated with the amplitudes of angry ERP ($R_{106} = 0.61$, $P < 0.001$) but not correlated with neutral ERP ($R_{106} = -0.18$, $P = 0.06$) (Fig. 4b).

**Sensitivity test results of MAOA in female participants**. Because men can only be classified by having *MAOA-high* or *MAOA-low* activity, whereas women can be classified as having *MAOA-high*, *MAOA-intermediate* or *MAOA-low*, we conducted three ANOVA models as to examine the effect of genetic variant on the EEG activity in female participants: (1) the first model comprising *MAOA-uVNTR* genotypes (High vs. Intermediate vs. Low) and *5-HTT* genotype (SS vs. LL/LS) as the between-subject factors; (2) the second model comprising *MAOA-uVNTR* genotypes (High/Intermediate vs. Low) and *5-HTT* genotype (SS vs. LL/LS) as the between-subject factors; (3) the third model comprising *MAOA-uVNTR* genotypes (High vs. Intermediate/Low) and *5-HTT* genotype (SS vs. LL/LS) as the between-subject factors.

For the fearful MMN, all the main effect of *MAOA* [model 1: $F_{2,72} = 0.335$, $P = .717$, $\eta p^2 = 0.009$, $(1-\beta) = 10.48\%$; model 2: $F_{1,74} = 0.663$, $P = 0.418$, $\eta p^2 = 0.009$, $(1-\beta) = 13.21\%$; model 3: $F_{1,74} < 0.001$, $P = 0.984$, $\eta p^2 < 0.001$, $(1-\beta) < 5\%$], main effect of *5-HTT* [model 1: $F_{2,72} = 0.217$, $P = 0.643$, $\eta p^2 = 0.003$, $(1-\beta) = 6.74\%$; model 2: $F_{1,74} = 0.768$, $P = 0.384$, $\eta p^2 = 0.01$, $(1-\beta) = 14.16\%$; model 3: $F_{1,74} = 0.042$, $P = 0.839$, $\eta p^2 = 0.001$, $(1-\beta) = 5.88\%$], and *MAOA × 5-HTT* interaction [model 1: $F_{2,72} = 0.499$, $P = 0.609$, $\eta p^2 = 0.014$, $(1-\beta) = 13.85\%$; model 2: $F_{1,74} = 0.399$, $P = .529$, $\eta p^2 = 0.005$, $(1-\beta) = 9.48\%$; model 3: $F_{1,74} = 0.31$, $P = 0.58$, $\eta p^2 = 0.004$, $(1-\beta) = 8.57\%$] was not significant in women.

As for the angry MMN, all the main effects of the *MAOA* [model 1: $F_{2,72} = 0.145$, $P = 0.865$, $\eta p^2 = 0.004$, $(1-\beta) = 7.33\%$; model 2: $F_{1,74} = 0025$, $P = 0.874$, $\eta p^2 < 0.001$, $(1-\beta) < 5\%$; model 3: $F_{1,74} = 0.282$, $P = 0.597$, $\eta p^2 = 0.004$, $(1-\beta) = 8.57\%$], main effect of *5-HTT* [model 1: $F_{2,72} = 3.106$, $P = 0.082$, $\eta p^2 = 0.041$, $(1-\beta) = 34.1\%$; model 2: $F_{1,74} = 3.901$, $P = 0.052$, $\eta p^2 = 0.05$, $(1-\beta) = 51.62\%$; model 3: $F_{1,74} = 2.062$, $P = 0.155$, $\eta p^2 = 0.027$, $(1-\beta) = 30.63\%$], and *MAOA × 5-HTT* interaction [model 1: $F_{2,72} = 0.231$, $P = 0.794$, $\eta p^2 = 0.006$, $(1-\beta) = 8.57\%$; model 2: $F_{1,74} = 0.288$, $P = 0.594$, $\eta p^2 = 0.004$, $(1-\beta) = 8.57\%$; model 3: $F_{1,74} = 0.082$, $P = 0.776$, $\eta p^2 = 0.001$, $(1-\beta) = 5.88\%$] was not significant in women. It is noteworthy however that although the effect size seems to be lower in women, the lack of findings in female participants may not be due to a lack of effect, but rather due to the genetic distribution that was not amenable enough as to detect any significance with the given sample size.

## Discussion

The main aim of this study was that of elucidating the relationships and interactions between the *5-HTTLPR*, the *MAOA-uVNTR*, and anxiety-related traits through the use of multimodal indices—including genetic, neurophysiological, and behavioral measurements. Contrary to our hypotheses, the findings revealed that the effects of the *MAOA-uVNTR* are not directly observed at the behavioral level (as assessed by the STAI). They are, however, capable to be observed on the fearful MMN amplitudes, albeit

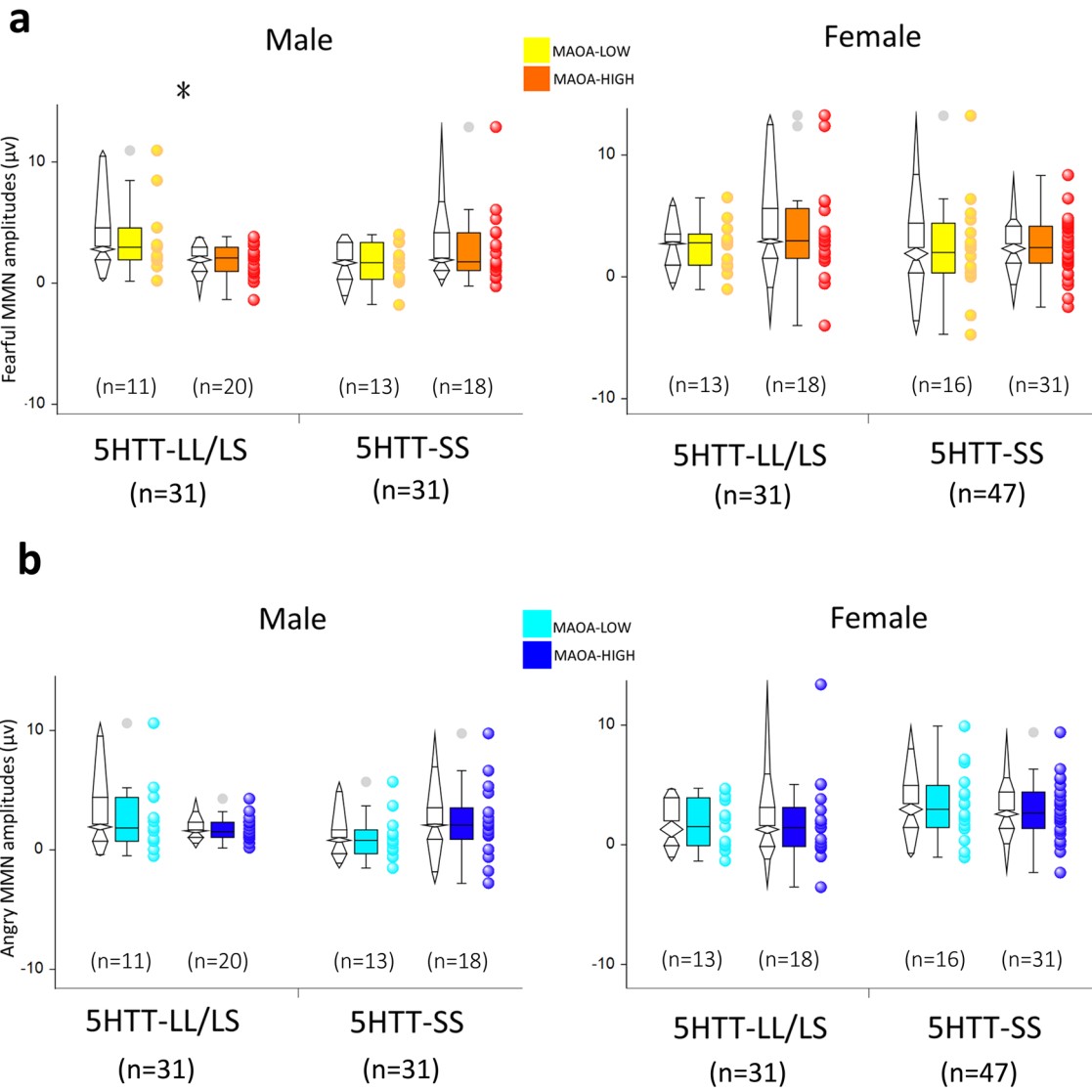

**Fig. 3 The interaction effects of MAOA-uVNTR and 5-HTT gene variants on the eMMN. a Fearful MMN**. The interaction effect of *MAOA-uVNTR × 5-HTT* was significant in male ($n = 62$) ($F_{1, 58} = 6.26$, $P = 0.015$, $\eta p^2 = 0.097$) but not in female participants ($n = 78$) ($F_{1, 74} = 0.399$, $P = 0.529$, $\eta p^2 = 0.005$). In men, MOAO exerted its effect only in ones who do not possess a 5-HTT genotype of SS ($n = 31$) [$T(29) = 2.33$, $P = .027$]. Men with *MAOA*-low (n = 11) ($3.74 \pm 0.97$ μV) showed significant larger fearful MMN than men with *MAOA*-high (n = 20) ($1.84 \pm 0.3$ μV) in the *5-HTT*-LL/LS group, whereas they are parallel in the *5-HTT*-SS group [*MAOA*-low (n = 13): $1.66 \pm 0.49$ μV; *MAOA*-high ($n = 18$): $2.88 \pm 0.72$ μV; $T(29) = -1.3$, $P = 0.2$].
**b Angry MMN**. The interaction effect of *MAOA-uVNTR × 5-HTT* was not significant in both male and female participants (male: $F_{1, 58} = 3.247$, $P = 0.077$, $\eta p^2 = 0.053$; female: $F_{1, 74} = 0.288$, $P = 0.593$, $\eta p^2 = 0.004$). Whisker boundaries were set as box edge ± 1.5 Inter-Quartile Range (IQR). Source data are presented in Supplementary Data 2.

only in the male subgroup. Furthermore, in said subgroup, the *MAOA-uVNTR* exerted its effect only in those men who were not homozygous for the S allele. That is, and contrary to our second hypothesis, that *MAOA*-L men had significant larger fearful MMN amplitudes than *MAOA*-H men in the *5-HTT*-LL/LS group, whereas fearful MMN amplitudes were not significantly different among the *MAOA*-L and *MAOA*-H men in the SS group. Interestingly, the significant larger fearful MMN amplitudes found among the *MAOA*-L men in the *5-HTT*-LL/LS group was due to increased ERPs in response to the fearful stimuli. Thus, there is a differential effect between the *MAOA-uVNTR* and the *5-HTTLPR*, with the latter eliciting smaller fearful MMN amplitudes as a function of increased ERPs in response to neutral stimuli. Further correlational analyses corroborated that the *MAOA*-L modulatory effect on the fearful MMN amplitudes was related to fearful ERPs but not to neutral ERPs.

It is no surprise that the effects of the *MAOA-uVNTR × 5-HTT* interaction on fearful MMN were only significant in men, as the *MAOA* enzyme is encoded in the X chromosome[41]. Thus, while men can be easily divided into 2 main groups (*MAOA*-L and *MAOA*-H), women have three levels of modulation (low, intermediate, and high), making *MAOA-uVNTR* assessments more complex when performed in the latter group.

On one hand, it is reasonable that the *MAOA*-L genotype exerts its influence only on the fearful MMN amplitudes of those men who possess *5-HTT* homozygous L alleles and those with *5-HTT* heterozygous LS alleles, as those homozygous for the S allele would have an increased bioavailability of serotonin in their synaptic clefts, while at the same time, the *MAOA*-L would fail to degrade serotonin effectively, thus incurring in a ceiling effect. Conversely, men possessing the *MAOA*-L genotype exhibited significant larger fearful MMN amplitudes than men with the *MAOA*-H phenotype

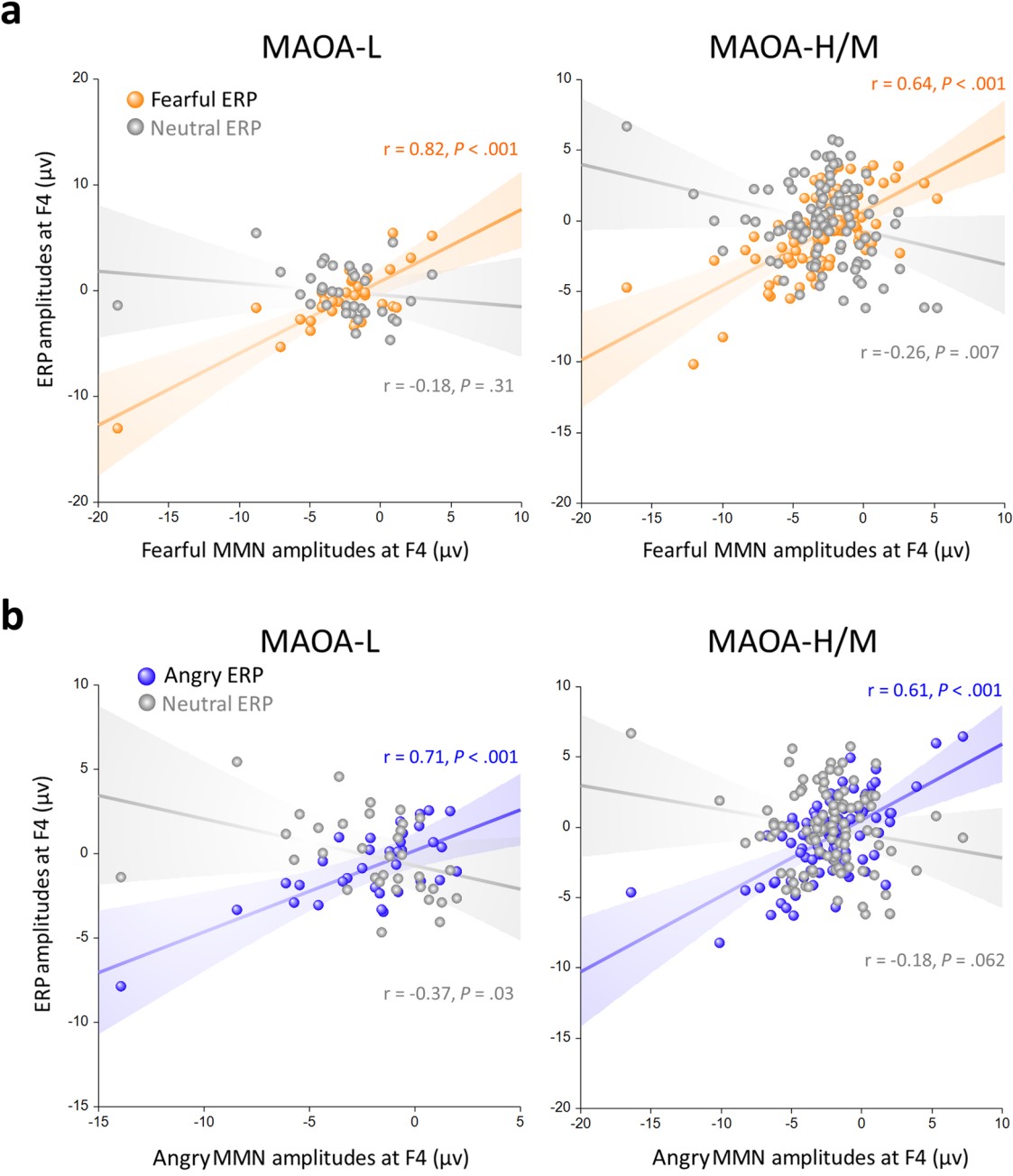

**Fig. 4 Emotional MMN as a function of neutral, angry and fearful ERPs in participants who possess different MAOA alleles. a Fearful MMN.** In participants with *MAOA*-low (combined men and women, $n = 34$, left panel), fearful MMN was only positively correlated with the amplitudes of fearful ERP ($R_{34} = 0.82$, $P < 0.001$) but not correlated with neutral ERP ($R_{34} = -0.18$, $P = 0.31$). However, in the rest of participants who have *MAOA*-H or *MAOA*-I (combined men and women, $n = 106$, right panel), fearful MMN was positively correlated with the amplitudes of fearful ERP ($R_{106} = 0.64$, $P < 0.001$) and negatively correlated with neutral ERP ($R_{106} = 0.26$, $P = 0.007$) **b. Angry MMN.** In participants with *MAOA*-low (combined men and women, $n = 34$), angry MMN was positively correlated with the amplitudes of angry ERP ($R_{34} = 0.71$, $P < 0.001$) and negatively correlated with neutral ERP ($R_{34} = -0.37$, $P = 0.03$). However, in the rest of participants who have *MAOA*-H or *MAOA*-I (combined men and women, $n = 106$), angry MMN was only positively correlated with the amplitudes of angry ERP ($R_{106} = 0.61$, $P < 0.001$) but not correlated with neutral ERP ($R_{106} = -0.18$, $P = 0.06$). Source data are presented in Supplementary Data 3.

in the *5-HTT*-LL/LS group, and with such significant difference being driven by an increase in ERP responses to fearful stimuli. This is probably due to the *5-HTT*-LL/LS genotypes dispelling the ceiling effect of which the *MAOA*-L men homozygous for the S allele are subject of, as one of the two neural mechanisms tasked with regulating serotonin—namely, that of serotonin reuptake—could still be effectively exerting some reduction in regard to serotonergic-receptor excitation.

On the other hand, it is imperative to remember that the mechanism through which those homozygous for the S allele incur in increased ERPs to neutral stimuli, as well as anxiety-related symptomatology, is that of amygdala hyperactivity as procured by the S allele[2,7], and which is not observed in those who are noncarriers. Although the amygdala has been seen to be paramount in both freezing and fight-or-flight responses to threat[42,43], it has been observed to be much more—and

significantly—associated with freezing responses and threat-induced motor inhibition, rather than with actual fighting actions[43]. As such, anxiety has been seen to be highly correlated with freezing behaviors[44,45]; hence the MAOA-uVNTR gene not exerting any effect over the STAI behavioral measure in our findings. Furthermore, and most importantly, the MAOA-uVNTR gene is not only in charge of the catabolism of serotonin, but also of the degradation of norepinephrine. The major source of nor-epinephrine and key structure of the brain's noradrenergic system is the locus coeruleus (LC)[46]. Furthermore, strong LC activation has been seen as a main factor by which network connectivity shifts in favor of salience processing in order to detect threat[47]. This might explain the reason why the increased fearful MMN amplitudes among men with the MAOA-L genotype are driven by ERPs as elicited by fearful cues, probably because these individuals may be more susceptible to negativity biases, the finding which is also in accordance with previous fMRI findings[17]. Moreover, norepinephrine has been observed to be central for enhancing arousal and promoting action[48,49], and subsequently enabling aggression. Furthermore, the impact of norepinephrine in aggressive behavior is not only at a single level, but rather ternary, as its effects are felt at (I) the hormonal level, where it appears to be involved in the metabolic preparations for a potential fight, (II) at the level of the sympathetic autonomous nervous system, where it ensures appropriate cardiovascular response, and (III) at the level of the central nervous system, where it readies the organism for a probable imminent fight[50].

In addition, although fear and anger are the two most widely investigated threat-related negative emotions that have been found to be able to affect human behaviors (even with the absence of conscious awareness)[31], these two emotions act along different motivational directions, with anger as an approach-motivated negative emotion, and fear as an avoidance-motivated negative emotion. Anger and fear were therefore found to assert various effects in the adaptability of the human defense system[51,52]. While individuals with MAOA-L mainly relied on the fearful ERPs—but not the neutral ERPs—during the preattentive processing of fearful MMN, they seemed to be more sensitive to and relying on both angry and neutral ERPs during the preattentive processing of angry MMN. In future studies, the hyperresponsive emotional arousal and the pronounced brain-volume reductions previously found in MAOA-L should be refined as to delineate the genotype effect on approach- and avoidance-motivated negative emotions[17].

Regarding the gender differences in emotion perception, although previous reviews showed a small-to-moderate female-gender-related advantage on emotional sensitivity, inconsistent evidence from recent studies has raised questions regarding the influence of different methodologies, stimuli, and samples[53]. In recent years, while our team used the same stimuli and paradigm on the investigation of emotional MMN, we were unable to report a stable gender effect consistently across our studies, with some findings suggesting female-gender related advantages yielding larger MMN amplitudes[54,55], albeit many of them with null results[7,24–26,56–58]. However, male gender-related advantages in emotional MMN have never been identified. The larger effect size regarding the male gender found in the current study might be underestimated due to the given sample size. Therefore, this male gender-related advantages—partially attributed to the MAOA-L genotype—cannot be fully ascribed to the fact that male participants perceived an opposite-gender voice, whereas female participants perceived a same-gender voice.

On the other hand, although the overexcitation to threatening emotions was found to be associated with both fearlessness and anxiety, the fearlessness results were mainly reported from men with MAOA-L[17,29], while the anxiety results were especially done on women with 5-HTTLPR short alleles[7,30]. The hyperactivity

found in male and female participants might be associated with various domains of the human defense system. The larger fearful MMN found in MAOA-L men might be associated with aggressive traits, whereas the larger angry MMN found in 5HTT-SS women might be more related to certain susceptibility regarding anxiety. However, due to the lack of measures regarding aggressive behavior, this refined hypothesis remains a future venue of inquiry.

It is important to note some potential limitations for the present study. First, the use of the pseudoword dada in the experimental design might impact the generalization of this research, as this might affect the proper representation of emotions. Nevertheless, other research using nonlinguistic emotional articulations verifies the utilized passive oddball paradigm as an optimal tool for emotional salience detection[7,59]. Second, the sample size might be small to effectively assess gene × behavior interactions. The lack of findings in female participants may not be due to a lack of effect, but rather due to the genetic distribution that was not amenable enough as to detect any significance with the given sample size. Notwithstanding, this study (a) uses a similar or greater sample size as that of other research assessing the same types of gene × behavior associations[7,17,60], and (b) this study makes use of endophenotypes, such that the relationship between the variables is of the order gene × brain × behavior, and which has been highly suggested by other researchers[2], as it provides a stronger association between genes and such complex phenomena as behaviors. Third, while there is a strong comorbidity between depression and anxiety disorders, the evidence in favor of dissociating subclinical anxiety from subthreshold depressive conditions is sparse[61,62]. However, without the assessments of depressive symptomatology, current research cannot yield professional benefits to this field. Future studies with both subclinical measures of anxiety and depressive symptoms are warranted. Finally, due to our original goal being that of exploring the endophenotypes for anxiety and moral attitudes, we did not assess aggressive behavior in none of the subjects used for this study, thus, a link between the MAOA-uVNTR gene and aggressive behavior remains for future enquiry. Nevertheless, and as stated before, there is ample neuroscientific literature evidencing that the low enzymatic activity of the MAOA-L genotype does incur in aggression[9,15].

All in all, by using multimodal indices—including genetic, neurophysiological, and behavioral measurements—this study demonstrates the intricate relationship that exists between the 5-HTT polymorphism, the MAOA-uVNTR genotypes, and the environment. With the interaction between the 5-HTT polymorphism and the MAOA-L genotype in male individuals having an impact on the processing of threatening stimuli and on social cognition as a whole. Furthermore, the differential threat-induced responses among individuals with varying 5-HTT pairs of alleles and MAOA-uVNTR enzymatic expressions may point toward the possibility of at least two anxiety subtypes, highly dependent on the dominance of either the serotonergic system or the noradrenergic system—as affected by the MAOA-L genotype's low enzymatic activity.

## Materials and methods

**Subjects.** This study was part of the social neuroscience project: from Genetic Heterogeneity and Brain Connectome to Social Neuroscience (YM104078E), carried out in the National Yang-Ming University, which investigates the individual difference in genetic heterogeneity, anxiety, moral attitudes, brain structure, and functions (2016/10/11~2018/09/30)[7,63,64]. During this period, we evaluated 567 adults [(male/female ratio—266/301), aged between 18 and 63 (25.66 ± 8.19)] who were having their first appointment in the Social Neuroscience Laboratory. The genetic and EEG part of this study comprised 140 adults (male/female ratio—62/78), aged between 18 and 46 (23.82 ± 4.07) years. All participants were Han Chinese and right-handed. They participated in the study after providing written informed consent and were screened for major psychiatric illnesses (e.g., general anxiety disorder) by the Structured Clinical Interview for DSM-IV Axis I Disorders (SCID-I) and excluded if there was evidence of comorbid neurological disorders

(e.g., dementia, seizures), history of head injury, and alcohol or substance abuse or dependence within the past five years. All participants were with normal hearing (pure-tone average thresholds <15 dB HL) at the time of testing. The subjects included in the data analysis were subdivided into subroups based on genotyping results of two candidate genes: *MAOA-uVNTR* (*MAOA*-high vs. *MAOA*-intermediate vs. *MAOA*-low) and *5-HTT* (SS vs. LL/LS). For *5-HTT* gene, participants possessing two copies of the S allele were included in the SS group ($n = 78$) and those who are homozygous for L or one copy of the L allele were included in the LS/LL group ($n = 62$), respectively. For *MAOA-uVNTR* gene, since it is located on the X chromosome, the results from men and women were analyzed separately. Women can be classified as having high (H, $n = 29$), intermediate (M, $n = 39$) or low (L, $n = 10$) *MAOA-uVNTR* activity, but men can only be classified by having high ($n = 38$) or low activity ($n = 24$). The genotype frequencies for females were L: 12.8%, M: 50%, and H: 37.2%; for males, they were L: 38.7% and H: 61.3%. This study was part of the social neuroscience project: from Genetic Heterogeneity and Brain Connectome to Social Neuroscience (with IRB number YM104078E). This study was approved by the Ethics Committee of the National Yang-Ming University Hospital, and conducted in accordance to the guidelines of the Declaration of Helsinki. A written informed consent was obtained from all the participants, as well as were given a monetary compensation at the end of the study.

**DNA extraction and genotyping**. Buccal cells were harvested from the inner cheek of each subject to provide DNA for genetic testing. The DNA was extracted from buccal swabs using a QIAamp DNA Mini Kit. The procedure employed a polymerase chain reaction (PCR)-based protocol followed by restriction endonuclease digestion to identify the *5-HTTLPR* located in the promoter region of the serotonin transporter gene (SLC6A4) and rs25531 variants: S, LA, and LG. Forward primer: 5′-TCCTCCGCTTTGGCGCCTCTTCC-3′ and reverse primer: 5′-TgggggTTgCAggggAgATCCT-3′ (10 μM each) were used for 50 μl of PCR containing about 25 ng of DNA, 25 μl Taq DNA Polymerase 2× Master Mix Red (Ampliqon), and ddH2O, with an initial 5-min denaturation step at 95 °C followed by 35 PCR cycles of 95 °C (30 s), 65 °C (40 s), and 72 °C (30 s), and a final extension step of 5 min at 72 °C. To distinguish the A/G single-nucleotide polymorphism of the rs25531, we extracted 10 ul of the PCR product for digestion by FastDigest HpaII (Thermo, FD0514), an isoschizomer of MspI, a total reaction of 20 ul. These were loaded side by side on 2.5–3.0% agarose gel. For detail, agarose-gel electrophoresis is conducted with the amplified PCR product and the samples after restriction endonuclease digestion. The *5-HTTLPR* amplicon length of S genotype is 469 bp, L is 512 bp. After the restriction digest, the fragment lengths of alleles: SA is 469 bp, SG is 402 bp and 67 bp, LA is 512 bp, L G is 402 bp, and 110 bp. Therefore, by the size difference of the PCR product, we can dissect the genotype of 5-HTTLPR.

For *MAOA-uVNTR* genotyping, PCR fragments were amplified using the forward primer: 5′-GCTGGTCTCTAAGAGTGGGTAC-3′ and reverse primer: 5′-GAACGGACGCTCCATTCGGAC-3 (10 μM each) were used for 50 μl PCR containing with 50 ng of DNA, 25 μl Taq DNA Polymerase 2× Master Mix Red (Ampliqon), and ddH2O, with an initial 5-min denaturation step at 95 °C followed by 35 PCR cycles of 95 °C (30 s), 68 °C (40 s), and 72 °C (30 s), and a final extension step of 5 min at 72 °C. These were loaded side by side on 2.5–3.0% agarose gel, with the amplified PCR product fragment lengths of alleles: 2.5 R (321 bps), 3.5 R (351 bps), 4.5 (381 bps), and 5.5 R (411 bps), we can dissect the genotype of *MAOA-uVNTR*.

**Stimuli**. The auditory stimuli for the ERP recordings were emotional syllables. A young female speaker produced the spoken syllables *dada* with fearful, angry, and neutral prosodies. Within each set of emotional syllables, the speaker produced the syllables for more than ten times[65]. Emotional syllables were edited to become equally long (550 ms) and loud (min: 57 dB, max: 62 dB; mean: 59 dB) using Sound Forge 9.0 and Cool Edit Pro 2.0. Each set was rated for emotionality on a 5-point Likert scale. Emotional syllables that were consistently identified as the extremely fearful and angry, as well as the most emotionless were selected as the fearful, angry, and neutral stimuli, respectively. The ratings on the Likert scale (mean ± SD) for the fearful, angry, and neutral syllables were 4.34 ± 0.65, 4.26 ± 0.85, and 2.47 ± 0.87, respectively [see[24–26,54–58,65–67] for validation].

**Procedures**. This study assessed state and trait anxiety (STAI) in one hundred and forty healthy volunteers, genotyped the 5-HTTLPR and MAOA-uVNTR genes, as well as recorded the eMMN. After recording ERPs, the State-Trait Anxiety inventory (STAI) was administered to the participants as to determine their self-reported anxiety levels[68]. State anxiety (STAI-S) indicates anxiety in specific situations, and trait anxiety (STAI-T) determines anxiety as a general trait. Given that scoring in the top range of the STAI-T suggests these participants might be experiencing some type of undiagnosed or previously unreported anxiety disorder, we used a structured clinical interview to ensure that none of the subjects had any evidence of such conditions.

**EEG apparatus and recordings**. The ERP recordings were conducted in an electrically shielded room. Stimuli were presented binaurally via two loudspeakers placed on the right and the left side of the subject's head. The sound-pressure level

(SPL) peaks of different types of stimuli were equalized to eliminate the effect of the angry stimuli's substantially greater energy. The mean background noise level was around 35-dB SPL. During recording, participants were required to watch a muted movie (doraemon cartoon) with subtitles, while task-irrelevant emotional syllables in oddball sequences were presented, as to control for attentional modulation. Participants were told to ignore the task-irrelevant emotional syllables. The passive oddball paradigm employed the fearful and angry syllables as deviants, and the neutral syllables as standards. There were two blocks. Each block consisted of 450 trials, of which 80% were neutral syllables, 10% were fearful syllables, and the other 10% were angry syllables. The sequences of stimuli were quasi-randomized such that successive deviant stimuli were avoided. The stimulus-onset asynchrony was 1200 ms, including a stimulus length of 550-ms and a 650-ms interstimulus interval.

The electroencephalogram was continuously recorded from 32 scalp sites using electrodes mounted in an elastic cap, and positioned according to the modified International 10–20 system, with the addition of two mastoid electrodes. The electrode at the right mastoid (A2) was used as the online reference. Eye blinks and eye movements were monitored with electrodes located above and below the left eye. The horizontal electro-oculogram was recorded from electrodes placed 1.5 cm lateral to the left and right external canthi. A ground electrode was placed on the forehead. Electrode/skin impedance were kept <5 kΩ. Channels were rereferenced offline to the average of left and right mastoid recordings [(A1 + A2)/2]. Signals were sampled at 500 Hz, band-pass filtered (0.1–100 Hz), and epoched over an analysis time of 900 ms, which included 100 ms of prestimulus used for baseline correction. An automatic artifact-rejection system excluded from the average all trials containing transients exceeding ±70 μV at recording electrodes [percentage of rejected trials (mean ± sd): neutral standard, 19 ± 17%; angry deviant: 20 ± 19%; fearful deviant: 17 ± 17%] and exceeding ±100μV at the ocular EOG (horizontal or vertical) channels [neutral standard, 31 ± 16%; angry deviant: 30 ± 16%; fearful deviant: 33 ± 17%]. The percentage of valid trials was parallel in each case ($P > 0.9$). Furthermore, the quality of ERP traces was ensured by careful visual inspection in every subject and trial, and by applying an appropriate digital, zero-phase-shift band-pass filter (0.1–50 Hz, 24 dB/octave). The first ten trials were omitted from the averaging in order to exclude unexpected large responses elicited by the initiation of the sequences. The paradigm was edited using the MatLab software (The MathWorks, Inc., USA). Each event in the paradigm was associated with a digital code that was sent to the continuous EEG, allowing offline segmentation and average of selected EEG periods for analysis. The ERPs were processed and analyzed using Neuroscan 4.3 (Compumedics Ltd., Australia).

**Statistical analyses**. The MMN amplitudes were defined as the average within a 50-ms time window surrounding the peak at the electrode sites F3, Fz, F4, C3, Cz, and C4. The peak was defined as the largest negativity of the difference between the deviant and standard ERPs during a period of 150–350 ms after stimulus onset. Only the standards before the deviants were included in the analysis. MMN was statistically tested using a mixed ANOVA comprising gender (male or female) as the between-subjects factor, and the deviant type (fearful or angry), coronal site (left, midline, and right) and anterior–posterior site (frontal or central) as the within-subjects factors. Degrees of freedom were corrected using the Greenhouse−Geisser method. A post hoc comparison was performed only when preceded by significant main effects. Statistical power (1-β) was estimated by G*Power 3.1 software[69]. Statistical analyses were performed using SPSS 17.0.

**Informed consent statement**. A written informed consent was obtained from all the participants, as well as were given a monetary compensation at the end of the study.

**Reporting summary**. Further information on research design is available in the Nature Research Reporting Summary linked to this article.

## Data availability
Data underlying Figs. 2–4 are presented in Supplementary Data 1–3, respectively. The raw data of PCR electrophoretic images and complete DNA analysis results are available from the corresponding author [CC], upon reasonable request.

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

## Acknowledgements

The study was funded by the Ministry of Science and Technology (MOST 108-2410-H-155-041-MY3; 110-2636-H-038-001-; 111-2636-H-038-008-; 110-2636-B-038-005-; 111-2636-B-038-004-).

## Author contributions

R.M.M., T.T.L., and C.C. conceived and conceptualized the study. R.M.M. and T.T.L. collected the data. R.M.M., T.T.L., Y.T.F., Y.C.C., and C.C. analyzed the data. R.M.M., T.T.L., and C.C. conducted the necessary literature reviews and drafted the first paper. R.M.M., T.T.L., Y.T.F., Y.C.C., and C.C. contributed toward the writing and revision of the final draft.

## Competing interests

The authors declare no competing interests.
