## [Transparent Peer Review File · Communications Biology]

Reviewers' comments:

Reviewer #2 (Remarks to the Author):

The authors present a very interesting and thorough work. They performed genetic, neurophysiological and cognitive measures on a large number of subjects to assess the interaction between the 5-HTTLPR and the MAOA-uVNTR genes and the responses to emotional stimuli as measured by eMMNs. Their work presented no relation amongst genotyping groups with STAI state or trait, which I found surprising. However they were able to associate a larger fearful response in males with the MAOA-low who were not homozygous to the 5HTT S allele. I believe this work contributes new relevant information to the field and would recommend its publication with a few minor revisions:

1) When evaluating eMMNs (Figure 2) both angry and fearful results are shown, where fearful presents a significantly larger eMMN amplitude. In the posterior results where the genetic influence is studied, only fearful data are analyzed and there is no mention to angry responses. I would appreciate a supplementary figure showing these results and a brief mention of the reason for which these are not included in the main work, even though I assume the reason is the result presented in Fig 2, this should be at least briefly addressed.

2) In the same line, I would also like to see the correlational analysis performed amongst neutral/fearful ERPs and eMMNs (Figure 4) for the angry/neutral data.

3) Regarding EEG processing authors state

"An automatic artifact rejection system excluded from the average all trials containing transients exceeding $\pm 70\mu V$ at recording electrodes and exceeding $\pm 100\mu V$ at the horizontal EOG channels" What was the percentage of rejected trials in each case?

4) Also regarding EEG processing I wonder if authors performed ICA rejection methods to exclude muscular or heartbeat artifacts.

5) Please avoid extremely long sentences such as

"To examine whether 5-HTT gene variants interact with the functional role of MAOA-uVNTR alleles that both encode proteins for central serotonergic functions, on the EEG activities in response to the pre-attentive processing of threatening voices, the fearful MMN mean amplitudes from midline and right site electrodes (FZ/CZ, F4/C4) that exhibited the largest ERPs were extracted and subjected into a two-way ANOVA comprising MAOA-uVNTR genotypes (High vs. Intermediate vs. Low) and 5HTT genotype (SS vs. LL/LS) as the between-subjects factors for men and women, respectively" Since they are hard to follow and points are lost in the word stream.

All that being said I would like to congratulate the authors on a very complete and profound experimental work and in their forth-coming approach to results contrary to their hypothesis.

Reviewer #3 (Remarks to the Author):

The present study is an examination of two genes, the serotonin transporter polymorphism (5-HTTLPR) and the monoamine oxidase A gene (MAOA-uVNTR), in relation to anxiety-related symptomatology and pre-attentive emotional processing, as measured by the emotional mismatch negativity (eMMN) event-related potential. The authors found that participants with non-homozygous 5-HTTLPR short alleles and low-activity MAOA-uVNTR displayed significantly larger eMMN amplitudes (i.e. fearful minus neutral), driven by larger responses to fearful stimuli, as compared to those with high-activity MAOA-uVNTR variants. This was only observed for male participants. The authors note the importance of this finding as it relates to the interaction of 5-HTT and MAOA-uVNTR variants and their effect on threat processing and social cognition in human males. Overall, this is an interesting and worth-while area of study, with a sophisticated methodology that combines genetic,

neurophysiological, and behavioural assessments. The methods are described in great detail and would allow for reproducibility. The findings are of great interest to others in the scientific community and to the wider field. However, there is some room for improvement in terms of the clarity of the rationale and the interpretation of the findings.

I have some specific questions/comments, as follows:

Abstract

1. It would be helpful if the number of participants was mentioned in the abstract.
2. 'eMMN' should be defined.

Introduction

3. Line 69 (p.3) – the paragraph is a run-on sentence and lacks clarity.
4. Line 133 (p.7) - The rationale surrounding fearful vs. threatening stimuli and anxiety vs. aggressive behaviour lacks clarity in the aims section. Is the purpose to examine the response to fearful stimuli as it related to anxiety symptoms or to aggressive behaviour?

Methodology

5. Line 458 (p.24) – A “young female speaker” was used for the auditory stimuli. What is the rationale for this? What effect might this have on the fact that significant effects were observed in male participants?
6. p.21 – Were any assessments of depressive symptomatology assessed in the participants (sub-clinical)? Given the strong co-morbidity with anxiety, and the relation with serotonin activity, controlling for depressive symptoms within the participants would be warranted.
7. Line 487 (p.25): “[...] participants were required to watch a muted movie with subtitles[...]” It would be helpful for the reader to know what type of movie was being presented (i.e. a movie with neutral content?)

Results

8. Line 164 (p.8) - A clarification of what is meant by “wave 1 data collection” would be helpful for the reader.
9. Line 165 (p.8) – The genotype groups were SS = 78 and LL/LS = 62; but above the groups are described as S/S, n = 80 and S/LG, LG/LG, S/LA, LG/LA, LA/LA = 29+6+12+10+3 = 60.
10. Line 232 (p. 12) – It was not clear to me why only the fearful MMN amplitudes were included in the examination of the effects of MAOA-uVNTR and 5-HTT genotypes on eMMN, particularly since the authors were seeking to evaluate “EEG activities in response to the pre-attentive processing of threatening voices”.
11. Line 235 (p.12) – If men and women were analyzed separately, would the MAOA-uVNTR genotype component of the two-way ANOVA be ‘High vs. Intermediate vs. Low’ for women but only ‘High vs. Low’ for men?
12. p. 43: Figure 2: It would be helpful to have the numbers in each group represented in the figure or within the figure caption.
13. p. 44: Figure 4: Should the y-axis read “ERP amplitudes” and not “ERP MMN amplitudes”?

Discussion

14. Line 291 (p.15) – It would be helpful to clarify that the lack of findings for females are not due to a lack of effects but due to the genetic distribution that was not amenable to detect results with the given sample size. The authors acknowledge the sample size limitations, however, for the female 5HTT LL group, there are only 8 participants, which is quite substantially low for conducting ANOVAs.
15. Line 293 (p. 15)- If women are divided into 3 groups, while men are divided into 2 groups, would

the solution be to recruit more females to enable higher ns in each group? Would this lead to a more pronounced difference between LL and HH in females?

16. Line 340 (p.19) – The relation of the current findings to theories of PTSD are beyond the scope of the current work, particularly the discussion of aggressive behaviour and PTSD. I would recommend shortening and/or removing this section of the discussion. Perhaps an elaboration of the approach/withdrawal hypothesis of emotion and the associated neural activity would be more relevant to the findings.

17. Line 375 (p.19) – The authors mention that they did not assess aggressive behavior in the participants; a justification of why they did not assess measures of aggression would be warranted, considering the numerous times aggressive behaviour (particular in males) is mentioned in the introduction and discussion.

RESPONSES TO THE REVIEWERS

COMMSBIO-21-1686A

We are very thankful to the Reviewers and the Editorial Team for taking their precious time in providing their constructive suggestions. The manuscript has been thoroughly revised (All revisions have been highlighted in the manuscript).

Reviewer #2

The authors present a very interesting and thorough work. They performed genetic, neurophysiological and cognitive measures on a large number of subjects to assess the interaction between the 5-HTTLPR and the MAOA-uVNTR genes and the responses to emotional stimuli as measured by eMMNs. Their work presented no relation amongst genotyping groups with STAI state or trait, which I found surprising. However, they were able to associate a larger fearful response in males with the MAOA-low who were not homozygous to the 5HTT S allele. I believe this work contributes new relevant information to the field and would recommend its publication with a few minor revisions:

- We thank the reviewer for their precious time and encouragement.
- We appreciate the reviewer giving us solid suggestions on the complement of angry MMN results. Furthermore, we have edited the manuscript thoroughly to avoid long sentences.

1) When evaluating eMMNs (Figure 2) both angry and fearful results are shown, where fearful presents a significantly larger eMMN amplitude. In the posterior results where the genetic influence is studied, only fearful data are analyzed and there is no mention to angry responses. I would appreciate a supplementary figure showing these results and a brief mention of the reason for which these are not included in the main work, even though I assume the reason is the result presented in Fig 2, this should be at least briefly addressed.

- Many thanks the reviewer for the thoughtful suggestion. We complemented the angry MMN analysis and rewrote the corresponding results and discussion to improve readability.
- New Figure 3 and new Figure 4 include the complementary results of angry MMN.
- **Results:** *Angry MMN*
“... As for the angry MMN, there was no main effect of MAOA-uVNTR [male: $F_{1,58} = 0.042$, $P = .838$, $\eta^2 = 0.001$, $(1-\beta) = 5.69\%$; female: $F_{1,74} = 0.025$, $P = .874$, $\eta^2 < 0.001$, $(1-\beta) < 5\%$], 5-HTT [male: $F_{1,58} = 0.84$, $P = .363$, $\eta^2 = 0.004$, $(1-\beta) = 7.81\%$; female: $F_{1,74} = 3.901$, $P = .052$, $\eta^2 = 0.05$, $(1-\beta) = 51.62\%$], nor MAOA-

uVNTR x 5-HTT interaction [male: $F_{1,58} = 3.247$, $P = .077$, $\eta p^2 = 0.053$, $(1-\beta) = 44.95\%$; female: $F_{1,74} = 0.288$, $P = .593$, $\eta p^2 = 0.004$, $(1-\beta) = 8.57\%$]. While the main effect of 5-HTT in women showed a marginal trend toward significance ($P = .052$), with a medium effect size, the planned pairwise comparison indicated that women with 5-HTT-SS tended to have a higher angry MMN amplitude than women with 5-HTT-LL/LS ($3.084 \pm 0.424 \mu V$, $1.788 \pm 0.501 \mu V$, $P = .052$, respectively).

Despite an interaction between MAOA-uVNTR and 5-HTT showing a marginal trend toward significance in men ($P = .077$), the planned pairwise comparisons did not reveal any significant simple main effect (all $P > .1$), albeit the MAOA effect in male participants dependent on the 5-HTT genotype in which men with MAOA-L showed an increased angry MMN in ones who don't possess a 5-HTT genotype of SS (MAOA-L: $2.72 \pm 0.95 \mu V$; MAOA-H: 1.76 ± 0.22 , $P = .34$) but a decreased angry MMN in ones with the 5-HTT-SS (MAOA-L: $1.08 \pm 0.53 \mu V$; MAOA-H: $2.29 \pm 0.7 \mu V$, $P = .18$) (Figure 3B).

Correlation analyses, conducted against ERPs and angry MMN amplitudes, showed that in participants with MAOA-L (combined men and women, $n = 34$), angry MMN was positively correlated with the amplitudes of angry ERP ($R_{34} = 0.71$, $P < .001$) and negatively correlated with neutral ERP ($R_{34} = -0.37$, $P = .03$). However, in the rest of participants who have MAOA-H or MAOA-I (combined men and women, $n = 106$), angry MMN was only positively correlated with the amplitudes of angry ERP ($R_{106} = 0.61$, $P < .001$) but not correlated with neutral ERP ($R_{106} = -0.18$, $P = .06$) (Figure 4B)."

➤ **Discussion:**

"...Additionally, although fear and anger are the two most widely investigated threat-related negative emotions that have been found to be able to affect human behaviors (even with the absence of conscious awareness)³¹, these two emotions act along different motivational directions, with anger as an approach-motivated negative emotion, and fear as an avoidance-motivated negative emotion. Anger and fear were therefore found to assert various effects in the adaptability of the human defense system^{51,52}. While individuals with MAOA-L mainly relied on the fearful ERPs –but not the neutral ERPs– during the pre-attentive processing of fearful MMN, they seemed to be more sensitive to and relying on both angry and neutral ERPs during the pre-attentive processing of angry MMN. In future studies, the hyperresponsive emotional arousal and the pronounced brain volume reductions previously found in MAOA-L should be refined so to delineate the genotype effect on approach- and avoidance-motivated negative emotions¹⁷..."

➤ **References:**

17. Tamietto, M. & de Gelder, B. Neural bases of the non-conscious perception of emotional signals. *Nat Rev Neurosci* 11, 697-709, doi:10.1038/nrn2889 (2010).
31. Meyer-Lindenberg, A. et al. Neural mechanisms of genetic risk for impulsivity and violence in humans. *Proc Natl Acad Sci U S A* 103, 6269-6274, doi:10.1073/pnas.0511311103 (2006).
51. Yin, H. et al. The Effects of Angry Expressions and Fearful Expressions on Duration Perception: An ERP Study. *Front Psychol* 12, 570497, doi:10.3389/fpsyg.2021.570497 (2021).
52. Habib, M., Cassotti, M., Moutier, S., Houde, O. & Borst, G. Fear and anger have opposite effects on risk seeking in the gain frame. *Front Psychol* 6, 253, doi:10.3389/fpsyg.2015.00253 (2015).

2) In the same line, I would also like to see the correlational analysis performed amongst neutral/fearful ERPs and eMMNs (Figure 4) for the angry/neutral data.

➤ Many thanks to the reviewer for the thoughtful suggestion. New Figure 4 includes the complementary results for the angry and neutral data.

➤ **New Figure 4:**

Emotional MMN as a function of neutral, angry and fearful ERPs in participants who possess different MAOA alleles

A. Fearful MMN. In participants with MAOA-L (combined men and women, $n = 34$, left panel), fearful MMN was only positively correlated with the amplitudes of fearful ERP ($R_{34} = 0.82$, $P < .001$) but not correlated with neutral ERP ($R_{34} = -0.18$, $P = .31$). However, in the rest of participants who have MAOA-H or MAOA-I (combined men and women, $n = 106$, right panel), fearful MMN was positively correlated with the amplitudes of fearful ERP ($R_{106} = 0.64$, $P < .001$) and negatively correlated with neutral ERP ($R_{106} = 0.26$, $P = .007$)

B. Angry MMN. In participants with MAOA-L (combined men and women, $n = 34$), angry MMN was positively correlated with the amplitudes of angry ERP ($R_{34} = 0.71$, $P < .001$) and negatively correlated with neutral ERP ($R_{34} = -0.37$, $P = .03$). However, in the rest of participants who have MAOA-H or MAOA-I (combined men and women, $n = 106$), angry MMN was only positively correlated with the amplitudes of angry ERP ($R_{106} = 0.61$, $P < .001$) but not correlated with neutral ERP ($R_{106} = -0.18$, $P = .06$).

3) Regarding EEG processing authors state

“An automatic artifact rejection system excluded from the average all trials containing transients exceeding $\pm 70\mu V$ at recording electrodes and exceeding $\pm 100\mu V$ at the horizontal EOG channels”

What was the percentage of rejected trials in each case?

- Many thanks to the reviewer for the thoughtful suggestion. We complemented the percentage of rejected trials in each case at each step, accordingly.
- **Methods:** EEG Apparatus and Recordings
“... An automatic artifact rejection system excluded from the average all trials containing transients exceeding $\pm 70\mu\text{V}$ at recording electrodes [percentage of rejected trials (mean \pm sd): neutral standard, $19\pm 17\%$; angry deviant: $20\pm 19\%$; fearful deviant: $17\pm 17\%$] and exceeding $\pm 100\mu\text{V}$ at the ocular EOG (horizontal or vertical) channels [neutral standard, $31\pm 16\%$; angry deviant: $30\pm 16\%$; fearful deviant: $33\pm 17\%$]. The percentage of valid trials was parallel in each case ($P > .9$).

4) Also regarding EEG processing I wonder if authors performed ICA rejection methods to exclude muscular or heartbeat artifacts.

- Many thanks to the reviewer for the thoughtful suggestion.
- Although many studies have proposed the use of ICA algorithms for artifact reduction, there appears to be a potential issue regarding ICA uncertainty, as its ability to introduce additional sources of variance remains significant (Pontifex et al., 2016); thus, we adopted a more conservative way of manual segment rejection to directly omit epochs that contaminated by artifacts without applying ICA transformation (Jiang et al., 2019). While artifact avoidance approaches like ICA could maintain more valid trials and hence increase signal power, this rejection method will significantly lose possibly useful neural signals.
- Because we have a relatively large trial number in each condition (with a minimum trial number > 80), we used stricter thresholds for rejecting trials compared with Guidelines for using ERPs to study human cognition (Picton et. al. 2000). Specifically, we excluded all trials containing transients exceeding $\pm 70\mu\text{V}$ at recording electrodes and exceeding $\pm 100\mu\text{V}$ at the ocular EOG (horizontal or vertical) channels.
- References:
X. Jiang, G.-B. Bian, and Z. Tian, “Removal of artifacts from eeg signals: a review,” *Sensors*, vol. 19, no. 5, p. 987, 2019.
M.B. Pontifex, K.L. Gwizdala, A.C. Parks, M. Billinger, C. Brunner Variability of ICA decomposition may impact EEG signals when used to remove eyeblink artifacts *Psychophysiology*, 54 (3) (2016), pp. 386-398, 10.1111/psyp.12804

T.W. Picton et. al. 2000. Guidelines for using event-related potentials to study cognition: Recording standards and publication criteria. *Psychophysiology* 37, 127-152.

5) Please avoid extremely long sentences such as

“To examine whether 5-HTT gene variants interact with the functional role of MAOA-uVNTR alleles that both encode proteins for central serotonergic functions, on the EEG activities in response to the pre-attentive processing of threatening voices, the fearful MMN mean amplitudes from midline and right site electrodes (FZ/CZ, F4/C4) that exhibited the largest ERPs were extracted and subjected into a two-way ANOVA comprising MAOA-uVNTR genotypes (High vs. Intermediate vs. Low) and 5HTT genotype (SS vs. LL/LS) as the between-subjects factors for men and women, respectively”

Since they are hard to follow and points are lost in the word stream.

- Many thanks to the reviewer for the thoughtful suggestion. This long sentence was edited and cut into several small sentences in order to improve the readability accordingly.
- **Results:** The effects of the MAOA-uVNTR and the 5-HTT genotypes on eMMN “... In order to examine whether the 5-HTT gene variants covariate with the functional role of the MAOA-uVNTR alleles, with both encoding proteins for central serotonergic functions, we further examined the interaction effect between the 5-HTT and the MAOA-uVNTR genotype on the EEG activities in response to the pre-attentive processing of threatening voices. Specifically, the mean amplitudes of fearful MMN and angry MMN from midline and right site electrodes (FZ/CZ, F4/C4) –where the largest ERPs were observed– were extracted and subjected into a two-way ANOVA. The MAOA-uVNTR genotypes (High vs. Low) and 5-HTT genotype (SS vs. LL/LS) were the between-subject factors. While MAOA is an X-linked gene, results were presented for men and women, respectively...”

All that being said I would like to congratulate the authors on a very complete and profound experimental work and in their forth-coming approach to results contrary to their hypothesis.

- Thank you very much! We appreciate the reviewer’s encouragement.

Reviewer #3:

The present study is an examination of two genes, the serotonin transporter polymorphism (5-HTTLPR) and the monoamine oxidase A gene (MAOA-uVNTR), in relation to anxiety-related symptomatology and pre-attentive emotional processing, as measured by the emotional mismatch negativity (eMMN) event-related potential. The authors found that participants with non-homozygous 5-HTTLPR short alleles and low-activity MAOA-uVNTR displayed significantly larger eMMN amplitudes (i.e. fearful minus neutral), driven by larger responses to fearful stimuli, as compared to those with high-activity MAOA-uVNTR variants. This was only observed for male participants. The authors note the importance of this finding as it relates to the interaction of 5-HTT and MAOA-uVNTR variants and their effect on threat processing and social cognition in human males. Overall, this is an interesting and worth-while area of study, with a sophisticated methodology that combines genetic, neurophysiological, and behavioural assessments. The methods are described in great detail and would allow for reproducibility. The findings are of great interest to others in the scientific community and to the wider field. However, there is some room for improvement in terms of the clarity of the rationale and the interpretation of the findings.

- We thank the reviewer for their precious time and solid suggestions.
- We carefully edited the manuscript and responded to the suggestions point-by-point accordingly.

I have some specific questions/comments, as follows:

Abstract

1. It would be helpful if the number of participants was mentioned in the abstract.

- Many thanks to the reviewer for the thoughtful suggestion. The number of participants was complemented in the abstract accordingly.
- **Abstract:**
“...Among the entire sample of participants in the study with valid genotyping and electroencephalographic (EEG) data (N=140), we show that men with low-activity MAOA-uVNTR, and who were not homozygous for the 5-HTTLPR short allele (s) (n=11), had significantly larger fearful MMN amplitudes –as driven by significant larger ERPs to fearful stimuli– than men with high-activity MAOA-uVNTR variants (n=20)...”

“...this study illustrates how the intricate interaction between the 5-HTT and the MAOA-uVNTR variants have an impact on threat processing, and social cognition, in male individuals (n=62).”

2. ‘eMMN’ should be defined.

- Many thanks to the reviewer for the thoughtful suggestion. In order to improve the readability, “eMMN” was replaced by “fearful MMN” accordingly.

Introduction

3. Line 69 (p.3) – the paragraph is a run-on sentence and lacks clarity.

- Many thanks to the reviewer for the thoughtful suggestion. This long sentence was reedited and burst into several small sentences in order to improve the readability accordingly.
- **Introduction:**
- “...Consequently, several neuroscientific studies delving into the research of the differential brain activations in S and L allele carriers, have observed that those carriers of the S allele incur in amygdala hyperactivity³. This over excitation not only has been associated to a susceptibility towards neuroticism or negative emotionality^{4,6}, but, furthermore, is the driving factor behind a mechanism encompassing an amygdala-related heightened baseline level of arousal even to non-threatening stimuli, and whose outcome appears to be anxiogenic symptomatology⁷...”
- References:
 - 3 Hariri, A. R. et al. Serotonin transporter genetic variation and the response of the human amygdala. *Science* 297, 400-403, doi:10.1126/science.1071829 (2002).
 - 4 Christou, A. I. et al. BDNF Val(66)Met and 5-HTTLPR Genotype are Each Associated with Visual Scanning Patterns of Faces in Young Children. *Front Behav Neurosci* 9, 175, doi:10.3389/fnbeh.2015.00175 (2015).
 - 5 Schinka, J. A., Busch, R. M. & Robichaux-Keene, N. A meta-analysis of the association between the serotonin transporter gene polymorphism (5-HTTLPR) and trait anxiety. *Mol Psychiatry* 9, 197-202, doi:10.1038/sj.mp.4001405 (2004).
 - 6 Sen, S., Burmeister, M. & Ghosh, D. Meta-analysis of the association between a serotonin transporter promoter polymorphism (5-HTTLPR) and anxiety-related personality traits. *Am J Med Genet B Neuropsychiatr Genet* 127B, 85-89, doi:10.1002/ajmg.b.20158 (2004).
 - 7 Chen, C. et al. An integrative analysis of 5-HTT-mediated mechanism of hyperactivity to non-threatening voices. *Commun Biol* 3, 113, doi:10.1038/s42003-020-0850-3 (2020).

4. Line 133 (p.7) - The rationale surrounding fearful vs. threatening stimuli and anxiety vs. aggressive behaviour lacks clarity in the aims section. Is the purpose to examine the response to fearful stimuli as it related to anxiety symptoms or to aggressive behaviour?

- Many thanks to the reviewer for the thoughtful suggestion.
- We carefully edited the part of the Introduction to delineate the rationale surrounding fearful vs. threatening stimuli and anxiety vs. aggressive behaviour in order to improve the readability accordingly.
- Albeit fear and anger are the two most commonly studied threat-related negative emotions that have been observed as being able to elicit amygdala activity even without conscious awareness, anger is an approach-motivated negative emotion, as opposed to fear, which instead is an avoidance-motivated negative emotion. Therefore, we further examined fearful and angry MMN separately.
- Regarding the relationship between anxiety and aggressive behaviour, the over excitation to threatening emotions was found to be associated with both fearlessness and anxiety. Nevertheless, the results for fearlessness were mainly reported among males with MAOA-L, while the results for anxiety were specific for women with 5-HTTLPR short alleles; thus, we speculate that the hyper-activity in MAOA-L males would be probably related to aggressive behavior, while the hyper-activity in women would perhaps relate to a susceptibility towards anxiety. However, due to the lack of data collected regarding aggressive behaviour, this field of knowledge remains for further query and out of the scope of this study. We carefully discussed this issue in the “Discussion” accordingly.
- **Introduction:** anxiety vs. aggressive behaviour
“...While the MAOA-L gene was found to be associated with a hyper-responsiveness to threatening stimuli and fearless temperament in men ^{17,29}, this over excitation to threatening emotions was also recognized as a pivotal bio-maker for 5-HTTLPR-related anxiogenic symptomatology ⁷ –condition particularly prevalent among the women population alongside depression and somatic³⁰. In order to examine the interaction effect of the two serotonin modulating genes, the 5-HTTLPR and MAOA-uVNTR, on the perception of threatening stimuli, and to test whether this genetic interaction co-varied with the factor gender, this study genotyped the 5-HTTLPR and MAOA-uVNTR, and recorded the MMN elicited by emotionally-spoken syllables in healthy male and female volunteers with varying degrees of state and trait anxiety...”
- **Introduction:** fearful vs. threatening stimuli
“...Meanwhile, while fear and anger are the two most commonly studied threat-related negative emotions that have been identified in previous studies as being able

to elicit amygdala activity even without conscious awareness³¹, these two emotions vary as a function of adaptability in terms of the human defense system. While anger is an approach-motivated negative emotion, fear acts as an avoidance-motivated negative emotion. We further examined the above-mentioned gender-gene interaction in electroencephalography (EEG) response to angry and fearful voices. Through the incorporation of multimodal indices –including genetic, neurophysiological, and behavioral measurements–, this study explores the possibility of a gene x gene x environment interaction as the starting point leading towards variation in social behavior...”

➤ **Discussion:**

“...Regarding the gender differences in emotion perception, although previous reviews showed a small to moderate female gender-related advantage on emotional sensitivity, inconsistent evidence from recent studies has raised questions regarding the influence of different methodologies, stimuli, and samples⁵³. In recent years, while our team used the same stimuli and paradigm on the investigation of emotional MMN, we were unable to report a stable gender effect consistently across our studies, with some findings suggesting female-gender related advantages yielding larger MMN amplitudes^{54,55}, albeit many of them with null results^{7,24-26,56-58}. However, male gender-related advantages in emotional MMN have never been identified. The larger effect size regarding the male gender found in the current study might be underestimated due to the given sample size. Therefore, this male gender-related advantages –partially attributed to the MAOA-L genotype– cannot be fully ascribed to the fact that male participants perceived an opposite-gender voice, whereas female participants perceived a same-gender voice. On the other hand, although the over excitation to threatening emotions was found to be associated with both fearlessness and anxiety, the fearlessness results were mainly reported from men with MAOA-L^{17,29}, while the anxiety results were especially done on women with 5-HTTLPR short alleles^{7,30}. The hyper-activity found in male and female participants might be associated with various domains of the human defense system. The larger fearful MMN found in MAOA-L men might be associated with aggressive traits, whereas the larger angry MMN found in 5-HTT-SS women might be more related to certain susceptibility regarding anxiety. However, due to the lack of measures for aggressive behavior, this refined hypothesis remains a future venue of inquiry.”

➤ **References:**

- 7 Chen, C. et al. An integrative analysis of 5-HTT-mediated mechanism of hyperactivity to non-threatening voices. *Commun Biol* 3, 113, doi:10.1038/s42003-020-0850-3 (2020).

- 24 Chen, C., Chan, C. W. & Cheng, Y. Test-Retest Reliability of Mismatch Negativity (MMN) to Emotional Voices. *Front Hum Neurosci* 12, 453, doi:10.3389/fnhum.2018.00453 (2018).
- 25 Chen, C., Hu, C. H. & Cheng, Y. Mismatch negativity (MMN) stands at the crossroads between explicit and implicit emotional processing. *Hum Brain Mapp* 38, 140-150, doi:10.1002/hbm.23349 (2017).
- 26 Chen, C., Lee, Y. H. & Cheng, Y. Anterior insular cortex activity to emotional salience of voices in a passive oddball paradigm. *Front Hum Neurosci* 8, 743, doi:10.3389/fnhum.2014.00743 (2014).
- 29 De Wied, M., Boxtel, A. V., Posthumus, J. A., Goudena, P. P. & Matthys, W. Facial EMG and heart rate responses to emotion-inducing film clips in boys with disruptive behavior disorders. *Psychophysiology* 46, 996-1004, doi:10.1111/j.1469-8986.2009.00851.x (2009).
- 30 Organization, W. H. Gender and women's mental health, <<https://www.who.int/teams/mental-health-and-substance-use/promotion-prevention/gender-and-women-s-mental-health>> (2021).
- 31 Tamietto, M. & de Gelder, B. Neural bases of the non-conscious perception of emotional signals. *Nat Rev Neurosci* 11, 697-709, doi:10.1038/nrn2889 (2010).
- 54 Hung, A. Y. & Cheng, Y. Sex differences in preattentive perception of emotional voices and acoustic attributes. *Neuroreport* 25, 464-469, doi:10.1097/WNR.000000000000115 (2014).
- 55 Fan, Y. T., Hsu, Y. Y. & Cheng, Y. Sex matters: n-back modulates emotional mismatch negativity. *Neuroreport* 24, 457-463, doi:10.1097/WNR.0b013e32836169b9 (2013).
- 56 Chen, C., Liu, C. C., Weng, P. Y. & Cheng, Y. Mismatch Negativity to Threatening Voices Associated with Positive Symptoms in Schizophrenia. *Front Hum Neurosci* 10, 362, doi:10.3389/fnhum.2016.00362 (2016).
- 57 Chen, C., Sung, J. Y. & Cheng, Y. Neural Dynamics of Emotional Salience Processing in Response to Voices during the Stages of Sleep. *Front Behav Neurosci* 10, 117, doi:10.3389/fnbeh.2016.00117 (2016).
- 58 Cheng, Y., Lee, S. Y., Chen, H. Y., Wang, P. Y. & Decety, J. Voice and emotion processing in the human neonatal brain. *J Cogn Neurosci* 24, 1411-1419, doi:10.1162/jocn_a_00214 (2012).

Methodology

5. Line 458 (p.24) – A “young female speaker” was used for the auditory stimuli. What is the rationale for this? What effect might this have on the fact that significant effects were observed in male participants?

- Many thanks to the reviewer for the thoughtful suggestion.
- Regarding to the gender differences in emotion perception and the effect that might be induced by the “same-gender voice” versus “different-gender voice” perceived by the participants, we have carefully complemented detailed discussion accordingly.
- **Discussion:**
- “...Regarding the gender differences in emotion perception, although previous reviews showed a small to moderate female advantage on emotional sensitivity, inconsistent evidence from recent studies has raised questions regarding the influence of different methodologies, stimuli, and samples⁵³. In recent years, while our team used the same stimuli and paradigm on the investigation of emotional MMN, we were unable to report a stable gender effect consistently across our studies, with some findings suggesting female-gender related advantages and larger MMN amplitudes^{54,55} albeit many of them with null results^{7,24-26,56-58}. However, male gender-related advantages in emotional MMN have never been identified. The larger effect size regarding the male gender found in the current study might be underestimated due to the given sample size. Therefore, this male gender-related advantages –partially attributed to the MAOA-L genotype– cannot be fully ascribed to the fact that male participants perceived an opposite-gender voice, whereas female participants perceived a same-gender voice.
On the other hand, although the over excitation to threatening emotions was found to be associated with both fearlessness and anxiety, the fearlessness results were mainly reported from men with MAOA-L^{17,29}, while the anxiety results were especially done for women with 5-HTTLPR short alleles^{7,30}. The hyper-activity found in male and female participants might be associated with various domains of the human defense system. The larger fearful MMN found in MAOA-L men might be associated with the aggressive traits, whereas the larger angry MMN found in 5-HTT-SS women might be more related to their susceptibility regarding anxiety. However, due to the lack of measures for aggressive behavior, this refined hypothesis remains a future interest of inquiry....”

6. p.21 – Were any assessments of depressive symptomatology assessed in the participants (sub-clinical)? Given the strong co-morbidity with anxiety, and the relation with serotonin activity, controlling for depressive symptoms within the participants would be warranted.

- Many thanks to the reviewer for the thoughtful suggestion. However, we did not measure the sub-clinical depressive symptomatology in our participants.

- We carefully discussed this study limitation regarding the co-morbidity between anxiety and depressive symptoms accordingly.
- **Discussion:**

“...Thirdly, while there is a strong comorbidity between depression and anxiety disorders, the evidence in favor of dissociating sub-clinical anxiety from subthreshold depressive conditions is sparse^{61,62}. However, without the assessments of depressive symptomatology, current research cannot yield professional benefits to this field. Future studies with both sub-clinical measures of anxiety and depressive symptoms are warranted...”
- References:
 - 61 Rodriguez, M. R., Nuevo, R., Chatterji, S. & Ayuso-Mateos, J. L. Definitions and factors associated with subthreshold depressive conditions: a systematic review. *BMC Psychiatry* 12, 181, doi:10.1186/1471-244X-12-181 (2012).
 - 62 Ng, J., Chan, H. Y. & Schlaghecken, F. Dissociating effects of subclinical anxiety and depression on cognitive control. *Adv Cogn Psychol* 8, 38-49, doi:10.2478/v10053-008-0100-6 (2012).

7. Line 487 (p.25): “[...] participants were required to watch a muted movie with subtitles[...].” It would be helpful for the reader to know what type of movie was being presented (i.e. a movie with neutral content?)

- Many thanks to the reviewer for the thoughtful suggestion.
- Yes, we presented a movie of “doraemon cartoon” with neutral content, while task-irrelevant emotional syllables in oddball sequences were presented.
- We edited the “Methods” section to have this information accordingly.
- **Methods:**

“...participants were required to watch a muted movie (doraemon cartoon) with subtitles...”

Results

8. Line 164 (p.8) - A clarification of what is meant by “wave 1 data collection” would be helpful for the reader.

- Many thanks to the reviewer for the thoughtful suggestion.
- Additional information regarding to the previous data collection was complemented accordingly.
- **Results:**

“...However, they had no significant deviation from the sample in wave 1 data collection, where the data was collected between the two calendar dates of

10/11/2016 and 07/02/2017, and was later published in March 2020 [$\chi^2(5) = 0.09$, $P = 0.99$]⁷”

➤ References:

7. Chen, C. et al. An integrative analysis of 5-HTT-mediated mechanism of hyperactivity to non-threatening voices. *Commun Biol* 3, 113, doi:10.1038/s42003-020-0850-3 (2020).

9. Line 165 (p.8) – The genotype groups were SS = 78 and LL/LS = 62; but above the groups are described as S/S, n = 80 and S/LG, LG/LG, S/LA, LG/LA, LA/LA = 29+6+12+10+3 = 60.

➤ We are deeply sorry for the mistake in the previous version of manuscript.

➤ After a through reexamination, the correct number was complemented accordingly.

➤ **Methods:**

➤ “...The 5-HTTLPR was found to have allele frequencies of S, n = 199 (71.1%); LA, n = 29 (10%); and LG, n = 52 (18.6%), and a genotype distribution of S/S, n = 78 (55.7%); S/LG, n = 30 (21.4%); LG/LG, n = 6 (4.3%); S/LA, n = 13 (9.3%); LG/LA, n = 10 (7.1%); and LA/LA, n = 3 (2.1%). Genotype distribution of the 5-HTTLPR of this sample was deviated from Hardy-Weinberg equilibrium, $\chi^2(3) = 10.602$, $P = 0.014$ partially due to the extremely low cases of LA/LA and the significantly higher S/S to L/L ratio which was previously identified in Han Chinese...”

10. Line 232 (p. 12) – It was not clear to me why only the fearful MMN amplitudes were included in the examination of the effects of MAOA-uVNTR and 5-HTT genotypes on eMMN, particularly since the authors were seeking to evaluate “EEG activities in response to the pre-attentive processing of threatening voices”.

➤ Many thanks to the reviewer for the thoughtful suggestion. We complemented the angry MMN analysis and rewrote the corresponding results and discussion to improve readability.

➤ New Figure 3 and new Figure 4 include the complementary results of angry MMN.

➤ **Results:** *Angry MMN*

“... As for the angry MMN, there was no main effect of MAOA-uVNTR [male: $F_{1, 58} = 0.042$, $P = .838$, $\eta p^2 = 0.001$, $(1-\beta) = 5.69\%$; female: $F_{1, 74} = 0.025$, $P = .874$, $\eta p^2 < 0.001$, $(1-\beta) < 5\%$], 5-HTT [male: $F_{1, 58} = 0.84$, $P = .363$, $\eta p^2 = 0.004$, $(1-\beta) = 7.81\%$; female: $F_{1, 74} = 3.901$, $P = .052$, $\eta p^2 = 0.05$, $(1-\beta) = 51.62\%$], nor MAOA-uVNTR x 5-HTT interaction [male: $F_{1, 58} = 3.247$, $P = .077$, $\eta p^2 = 0.053$, $(1-\beta) = 44.95\%$; female: $F_{1, 74} = 0.288$, $P = .593$, $\eta p^2 = 0.004$, $(1-\beta) = 8.57\%$]. While the main effect of 5-HTT in women showed a marginal trend toward significance (P

= .052), with a medium effect size, the planned pairwise comparison indicated that women with 5-HTT-SS tended to have a higher angry MMN amplitude than women with 5-HTT-LL/LS ($3.084 \pm 0.424 \mu\text{V}$, $1.788 \pm 0.501 \mu\text{V}$, $P = .052$, respectively).

Despite the interaction between the MAOA-uVNTR and 5-HTT genes showing a marginal trend toward significance in men ($P = .077$), the planned pairwise comparisons did not reveal any significant simple main effect (all $P > .1$); albeit the MAOA effect in male participants dependent on the 5-HTT genotype showing that men with MAOA-L yielded increased angry MMN among those men who do not possess a 5-HTT genotype of SS (MAOA-L: $2.72 \pm 0.95 \mu\text{V}$; MAOA-H: 1.76 ± 0.22 , $P = .34$), but decreased angry MMN in those with the 5-HTT-SS (MAOA-L: $1.08 \pm 0.53 \mu\text{V}$; MAOA-H: $2.29 \pm 0.7 \mu\text{V}$, $P = .18$) (Figure 3B).

Correlation analyses, conducted against ERPs and angry MMN amplitudes, showed that in participants with MAOA-L (combined men and women, $n = 34$), angry MMN was positively correlated with the amplitudes of angry ERP ($R_{34} = 0.71$, $P < .001$) and negatively correlated with neutral ERP ($R_{34} = -0.37$, $P = .03$). However, in the rest of participants who have MAOA-H or MAOA-I (combined men and women, $n = 106$), angry MMN was only positively correlated with the amplitudes of angry ERP ($R_{106} = 0.61$, $P < .001$) but not correlated with neutral ERP ($R_{106} = -0.18$, $P = .06$) (Figure 4B)."

➤ **Discussion:**

"...Additionally, although fear and anger are the two most widely investigated threat-related negative emotions that have been found to be able to affect human behaviors (even with the absence of conscious awareness)³¹, these two act along different motivational directions, with anger as an approach-motivated negative emotion, and fear as an avoidance-motivated negative emotion. Anger and fear were therefore found to assert various effects in the adaptability of the human defense system^{51,52}. While individuals with MAOA-L mainly relied on the fearful ERPs –but not the neutral ERPs– during the pre-attentive processing of fearful MMN, they seemed to be more sensitive to and relying on both angry and neutral ERPs during the pre-attentive processing of angry MMN. In the future studies, the hyperresponsive emotional arousal and the pronounced brain volume reductions previously found in MAOA-L should be refined so to delineate the genotype effect on approach- and avoidance-motivated negative emotions, respectively¹⁷..."

➤ **References:**

17. Tamietto, M. & de Gelder, B. Neural bases of the non-conscious perception of emotional signals. *Nat Rev Neurosci* 11, 697-709, doi:10.1038/nrn2889 (2010).
31. Meyer-Lindenberg, A. et al. Neural mechanisms of genetic risk for impulsivity and

violence in humans. *Proc Natl Acad Sci U S A* 103, 6269-6274,
doi:10.1073/pnas.0511311103 (2006).

51. Yin, H. et al. The Effects of Angry Expressions and Fearful Expressions on Duration Perception: An ERP Study. *Front Psychol* 12, 570497,
doi:10.3389/fpsyg.2021.570497 (2021).
52. Habib, M., Cassotti, M., Moutier, S., Houde, O. & Borst, G. Fear and anger have opposite effects on risk seeking in the gain frame. *Front Psychol* 6, 253,
doi:10.3389/fpsyg.2015.00253 (2015).

11. Line 235 (p.12) – If men and women were analyzed separately, would the MAOA-uVNTR genotype component of the two-way ANOVA be ‘High vs. Intermediate vs. Low’ for women but only ‘High vs. Low’ for men?

- Many thanks to the reviewer for the thoughtful suggestion. Because the effect of the MAOA-intermediate in female participants remains elusive, we examined MAOA-uVNTR genotypes in women in an explorative manner accordingly.
- We also complement the results with this explorative approach in the supplementary results.
- **Results:**
“Due to the effect of the MAOA-intermediate in female participants remaining elusive, we examined the MAOA-uVNTR genotypes in women in an explorative manner. Specifically, we tested the MAOA effect by treating it as a three-level variable (MAOA-high vs. MAOA-intermediate vs. MAOA-low), a two-level variable by regrouping MAOA-high and MAOA-intermediate (MAOA-high/intermediate vs. MAOA-low), or a two-level variable by regrouping MAOA-intermediate and MAOA-low (MAOA-high vs. MAOA-intermediate/low). Because the pattern of results was similar across the above-mentioned three models, we presented the results here for the MAOA-high/intermediate vs. MAOA-low model of our female participants, with the purpose of group size equalization (for full details on explorative analyses of MAOA in women, see Supplementary Results).”
- **Supplementary Results:**
“Because men can only be classified by having MAOA-high or MAOA-low activity, but women can be classified as having MAOA-high, MAOA-intermediate or MAOA-low, we conducted three ANOVA models to examine the effect of genetic variant on the EEG activity in female participants: (1) the first model comprising MAOA-uVNTR genotypes (High vs. Intermediate vs. Low) and 5-HTT genotype (SS vs. LL/LS) as the between-subjects factors; (2) the second model comprising MAOA-uVNTR genotypes (High/Intermediate vs. Low) and 5-HTT genotype (SS vs. LL/LS) as the between-subjects factors; (3) the third model comprising MAOA-

uVNTR genotypes (High vs. Intermediate/Low) and 5-HTT genotype (SS vs. LL/LS) as the between-subjects factors.

For the fearful MMN, all the main effect of MAOA [model 1: $F(2, 72) = 0.335$, $P = .717$, $\eta^2 = 0.009$, $(1-\beta) = 10.48\%$; model 2: $F(1, 74) = 0.663$, $P = .418$, $\eta^2 = 0.009$, $(1-\beta) = 13.21\%$; model 3: $F(1, 74) < 0.001$, $P = .984$, $\eta^2 < 0.001$, $(1-\beta) < 5\%$], main effect of 5-HTT [model 1: $F(2, 72) = 0.217$, $P = .643$, $\eta^2 = 0.003$, $(1-\beta) = 6.74\%$; model 2: $F(1, 74) = 0.768$, $P = .384$, $\eta^2 = 0.01$, $(1-\beta) = 14.16\%$; model 3: $F(1, 74) = 0.042$, $P = .839$, $\eta^2 = 0.001$, $(1-\beta) = 5.88\%$], and MAOA x 5-HTT interaction [model 1: $F(2, 72) = 0.499$, $P = .609$, $\eta^2 = 0.014$, $(1-\beta) = 13.85\%$; model 2: $F(1, 74) = 0.399$, $P = .529$, $\eta^2 = 0.005$, $(1-\beta) = 9.48\%$; model 3: $F(1, 74) = 0.31$, $P = .58$, $\eta^2 = 0.004$, $(1-\beta) = 8.57\%$] was not significant in women.

As for the angry MMN, all the main effect of MAOA [model 1: $F(2, 72) = 0.145$, $P = .865$, $\eta^2 = 0.004$, $(1-\beta) = 7.33\%$; model 2: $F(1, 74) = 0.025$, $P = .874$, $\eta^2 < 0.001$, $(1-\beta) < 5\%$; model 3: $F(1, 74) = 0.282$, $P = .597$, $\eta^2 = 0.004$, $(1-\beta) = 8.57\%$], main effect of 5-HTT [model 1: $F(2, 72) = 3.106$, $P = .082$, $\eta^2 = 0.041$, $(1-\beta) = 34.1\%$; model 2: $F(1, 74) = 3.901$, $P = .052$, $\eta^2 = 0.05$, $(1-\beta) = 51.62\%$; model 3: $F(1, 74) = 2.062$, $P = .155$, $\eta^2 = 0.027$, $(1-\beta) = 30.63\%$], and MAOA x 5-HTT interaction [model 1: $F(2, 72) = 0.231$, $P = .794$, $\eta^2 = 0.006$, $(1-\beta) = 8.57\%$; model 2: $F(1, 74) = 0.288$, $P = .594$, $\eta^2 = 0.004$, $(1-\beta) = 8.57\%$; model 3: $F(1, 74) = 0.082$, $P = .776$, $\eta^2 = 0.001$, $(1-\beta) = 5.88\%$] was not significant in women. It is noteworthy however that, although the effect size seems to be lower in women, the lack of findings in female participants may not be due to a lack of effect, but rather due to the genetic distribution that was not amenable enough as to detect any significance with the given sample size.”

12. p. 43: Figure 2: It would be helpful to have the numbers in each group represented in the figure or within the figure caption.

- Many thanks to the reviewer for the thoughtful suggestion.
- A new Figure 3 with the numbers in each group and the new results for angry MMN were complemented accordingly.

13. p. 44: Figure 4: Should the y-axis read “ERP amplitudes” and not “ERP MMN amplitudes”?

- Many thanks to the reviewer for the thoughtful suggestion. The y-axis of Figure 4 was corrected and the new results for angry MMN were complemented in the new Figure 4 accordingly.

Discussion

14. Line 291 (p.15) – It would be helpful to clarify that the lack of findings for females are not due to a lack of effects but due to the genetic distribution that was not amenable to detect results with the given sample size. The authors acknowledge the sample size limitations, however, for the female 5HTT LL group, there are only 8 participants, which is quite substantially low for conducting ANOVAs.

- Many thanks to the reviewer for the thoughtful suggestion. We have now explicitly stated that the lack of findings for females may due to the given small sample size accordingly.
- The smallest sample size among our ANOVAs cells is the one in men with low-activity MAOA-uVNTR, and who were not homozygous for the 5-HTTLPR short allele (s) (N=11).
- **Discussion:**
“...The lack of findings in female participants may not be due to a lack of effect, but rather due to the genetic distribution that was not amenable enough as to detect any significance with the given sample size...”

15. Line 293 (p. 15)- If women are divided into 3 groups, while men are divided into 2 groups, would the solution be to recruit more females to enable higher ns in each group? Would this lead to a more pronounced difference between LL and HH in females?

- Many thanks to the reviewer for the thoughtful suggestion. Because the effect of MAOA-intermediate in female participants remained elusive, we examined MAOA-uVNTR genotypes in women in an explorative manner accordingly.
- However, although we tried to collect and analyze a n=20 newly collected buccal cell dataset before EEG collection in the past month, we failed to get any new female data with the combination of 5-HTT-LL and MAOA-HH. Due to the inconvenience caused by the pandemic and the extremely low case of LA/LA in the Han Chinese population⁷, we complement the results with the explorative approach, as to try to compensate the sample size limitation.
- **Results:**
“Because the effect of MAOA-intermediate in female participants remains elusive, we examined MAOA-uVNTR genotypes in women in an explorative manner. Specifically, we tested the MAOA effect by treating it as a three-level variable (MAOA-high vs. MAOA-intermediate vs. MAOA-low), a two-level variable by regrouping MAOA-high and MAOA-intermediate (MAOA-high/intermediate vs. MAOA-low), or a two-level variable by regrouping MAOA-intermediate and MAOA-low (MAOA-high vs. MAOA-intermediate/low). Because the pattern of results was similar across the above-mentioned three models, we presented the results here for MAOA-high/intermediate vs. MAOA-low for our female

participants, with the purpose of group size equalization (for full details on explorative analyses of MAOA in women, see Supplementary Results)...”

➤ **Supplementary Results:**

“Because men can only be classified by having MAOA-high or MAOA-low activity, whereas women can be classified as having MAOA-high, MAOA-intermediate or MAOA-low, we conducted three ANOVA models to examine the effect of genetic variant on the EEG activity in female participants: (1) the first model comprising MAOA-uVNTR genotypes (High vs. Intermediate vs. Low) and 5-HTT genotype (SS vs. LL/LS) as the between-subject factors; (2) the second model comprising MAOA-uVNTR genotypes (High/Intermediate vs. Low) and 5-HTT genotype (SS vs. LL/LS) as the between-subject factors; (3) the third model comprising MAOA-uVNTR genotypes (High vs. Intermediate/Low) and 5-HTT genotype (SS vs. LL/LS) as the between-subject factors.

For the fearful MMN, all the main effect of MAOA [model 1: $F_{2, 72} = 0.335$, $P = .717$, $\eta^2 = 0.009$, $(1-\beta) = 10.48\%$; model 2: $F_{1, 74} = 0.663$, $P = .418$, $\eta^2 = 0.009$, $(1-\beta) = 13.21\%$; model 3: $F_{1, 74} < 0.001$, $P = .984$, $\eta^2 < 0.001$, $(1-\beta) < 5\%$], main effect of 5-HTT [model 1: $F_{2, 72} = 0.217$, $P = .643$, $\eta^2 = 0.003$, $(1-\beta) = 6.74\%$; model 2: $F_{1, 74} = 0.768$, $P = .384$, $\eta^2 = 0.01$, $(1-\beta) = 14.16\%$; model 3: $F_{1, 74} = 0.042$, $P = .839$, $\eta^2 = 0.001$, $(1-\beta) = 5.88\%$], and MAOA x 5-HTT interaction [model 1: $F_{2, 72} = 0.499$, $P = .609$, $\eta^2 = 0.014$, $(1-\beta) = 13.85\%$; model 2: $F_{1, 74} = 0.399$, $P = .529$, $\eta^2 = 0.005$, $(1-\beta) = 9.48\%$; model 3: $F_{1, 74} = 0.31$, $P = .58$, $\eta^2 = 0.004$, $(1-\beta) = 8.57\%$] was not significant in women.

As for the angry MMN, all the main effect of the MAOA [model 1: $F_{2, 72} = 0.145$, $P = .865$, $\eta^2 = 0.004$, $(1-\beta) = 7.33\%$; model 2: $F_{1, 74} = 0.025$, $P = .874$, $\eta^2 < 0.001$, $(1-\beta) < 5\%$; model 3: $F_{1, 74} = 0.282$, $P = .597$, $\eta^2 = 0.004$, $(1-\beta) = 8.57\%$], main effect of 5-HTT [model 1: $F_{2, 72} = 3.106$, $P = .082$, $\eta^2 = 0.041$, $(1-\beta) = 34.1\%$; model 2: $F_{1, 74} = 3.901$, $P = .052$, $\eta^2 = 0.05$, $(1-\beta) = 51.62\%$; model 3: $F_{1, 74} = 2.062$, $P = .155$, $\eta^2 = 0.027$, $(1-\beta) = 30.63\%$], and MAOA x 5-HTT interaction [model 1: $F_{2, 72} = 0.231$, $P = .794$, $\eta^2 = 0.006$, $(1-\beta) = 8.57\%$; model 2: $F_{1, 74} = 0.288$, $P = .594$, $\eta^2 = 0.004$, $(1-\beta) = 8.57\%$; model 3: $F_{1, 74} = 0.082$, $P = .776$, $\eta^2 = 0.001$, $(1-\beta) = 5.88\%$] was not significant in women. It is noteworthy however that, although the effect size seems to be lower in women, the lack of findings in female participants may not be due to a lack of effect, but rather due to the genetic distribution that was not amenable as to detect results with the given sample size.”

➤ **References:**

7. Chen, C. et al. An integrative analysis of 5-HTT-mediated mechanism of hyperactivity to non-threatening voices. *Commun Biol* 3, 113, doi:10.1038/s42003-

16. Line 340 (p.19) – The relation of the current findings to theories of PTSD are beyond the scope of the current work, particularly the discussion of aggressive behaviour and PTSD. I would recommend shortening and/or removing this section of the discussion. Perhaps an elaboration of the approach/withdrawal hypothesis of emotion and the associated neural activity would be more relevant to the findings.

- Many thanks to the reviewer for the thoughtful suggestion. The unrelated discussion of PTSD was removed and replaced by a detailed discussion on the rationale surrounding fearful vs. threatening stimuli, anxiety vs. aggressive behaviour, and gender differences, accordingly.
- **Discussion:**
- “...Additionally, although fear and anger are the two most widely investigated threat-related negative emotions that have been found to be able to affect human behaviors (even with the absence of conscious awareness)³¹, these two act along different motivational directions, with anger as an approach-motivated negative emotion, and fear as an avoidance-motivated negative emotion. Anger and fear were therefore found to assert various effects in the adaptability of the human defense system^{51,52}. While individuals with MAOA-L mainly relied on the fearful ERPs –but not the neutral ERPs– during the pre-attentive processing of fearful MMN, they seemed to be more sensitive to and relying on both angry and neutral ERPs during the pre-attentive processing of angry MMN. In the future studies, the hyperresponsive emotional arousal and the pronounced brain volume reductions previously found in MAOA-L should be refined so to delineate the genotype effect on approach- and avoidance-motivated negative emotions, respectively¹⁷. “Regarding the gender differences in emotion perception, although previous reviews showed a small to moderate female advantage on emotional sensitivity, inconsistent evidence from recent studies has raised questions regarding the influence of different methodologies, stimuli, and samples⁵³. In recent years, while our team used the same stimuli and paradigm on the investigation of emotional MMN, we were unable to report a stable gender effect consistently across our studies, with some findings suggesting female-gender related advantages and larger MMN amplitudes^{54,55} albeit many of them with null results^{7,24-26,56-58}. However, male gender-related advantages in emotional MMN have never been identified. The larger effect size regarding the male gender found in the current study might be underestimated due to the given sample size. Therefore, this male gender-related advantages –partially attributed to the MAOA-L genotype– cannot be fully ascribed

to the fact that male participants perceived an opposite-gender voice, whereas female participants perceived a same-gender voice.

“On the other hand, although the over excitation to threatening emotions was found to be associated with both fearlessness and anxiety, the fearlessness results were mainly reported from men with MAOA-L^{17,29}, while the anxiety results were especially done for women with 5-HTTLPR short alleles^{7,30}. The hyper-activity found in male and female participants might be associated with various domains of the human defense system. The larger fearful MMN found in MAOA-L men might be associated with the aggressive traits, whereas the larger angry MMN found in 5-HTT-SS women might be more related to their susceptibility regarding anxiety. However, due to the lack of measures for aggressive behavior, this refined hypothesis remains a future interest of inquiry....”

➤ References:

- 7 Chen, C. et al. An integrative analysis of 5-HTT-mediated mechanism of hyperactivity to non-threatening voices. *Commun Biol* 3, 113, doi:10.1038/s42003-020-0850-3 (2020).
- 17 Meyer-Lindenberg, A. et al. Neural mechanisms of genetic risk for impulsivity and violence in humans. *Proc Natl Acad Sci U S A* 103, 6269-6274, doi:10.1073/pnas.0511311103 (2006).
- 24 Chen, C., Chan, C. W. & Cheng, Y. Test-Retest Reliability of Mismatch Negativity (MMN) to Emotional Voices. *Front Hum Neurosci* 12, 453, doi:10.3389/fnhum.2018.00453 (2018).
- 25 Chen, C., Hu, C. H. & Cheng, Y. Mismatch negativity (MMN) stands at the crossroads between explicit and implicit emotional processing. *Hum Brain Mapp* 38, 140-150, doi:10.1002/hbm.23349 (2017).
- 26 Chen, C., Lee, Y. H. & Cheng, Y. Anterior insular cortex activity to emotional salience of voices in a passive oddball paradigm. *Front Hum Neurosci* 8, 743, doi:10.3389/fnhum.2014.00743 (2014).
- 29 De Wied, M., Boxtel, A. V., Posthumus, J. A., Goudena, P. P. & Matthys, W. Facial EMG and heart rate responses to emotion-inducing film clips in boys with disruptive behavior disorders. *Psychophysiology* 46, 996-1004, doi:10.1111/j.1469-8986.2009.00851.x (2009).
- 30 Organization, W. H. Gender and women's mental health, <<https://www.who.int/teams/mental-health-and-substance-use/promotion-prevention/gender-and-women-s-mental-health>> (2021).
- 31 Tamietto, M. & de Gelder, B. Neural bases of the non-conscious perception of emotional signals. *Nat Rev Neurosci* 11, 697-709, doi:10.1038/nrn2889 (2010).
- 51 Yin, H. et al. The Effects of Angry Expressions and Fearful Expressions on

- Duration Perception: An ERP Study. *Front Psychol* 12, 570497, doi:10.3389/fpsyg.2021.570497 (2021).
- 52 Habib, M., Cassotti, M., Moutier, S., Houde, O. & Borst, G. Fear and anger have opposite effects on risk seeking in the gain frame. *Front Psychol* 6, 253, doi:10.3389/fpsyg.2015.00253 (2015).
- 53 Fischer, A. H., Kret, M. E. & Broekens, J. Gender differences in emotion perception and self-reported emotional intelligence: A test of the emotion sensitivity hypothesis. *PLoS One* 13, e0190712, doi:10.1371/journal.pone.0190712 (2018).
- 54 Hung, A. Y. & Cheng, Y. Sex differences in preattentive perception of emotional voices and acoustic attributes. *Neuroreport* 25, 464-469, doi:10.1097/WNR.000000000000115 (2014).
- 55 Fan, Y. T., Hsu, Y. Y. & Cheng, Y. Sex matters: n-back modulates emotional mismatch negativity. *Neuroreport* 24, 457-463, doi:10.1097/WNR.0b013e32836169b9 (2013).
- 56 Chen, C., Liu, C. C., Weng, P. Y. & Cheng, Y. Mismatch Negativity to Threatening Voices Associated with Positive Symptoms in Schizophrenia. *Front Hum Neurosci* 10, 362, doi:10.3389/fnhum.2016.00362 (2016).
- 57 Chen, C., Sung, J. Y. & Cheng, Y. Neural Dynamics of Emotional Salience Processing in Response to Voices during the Stages of Sleep. *Front Behav Neurosci* 10, 117, doi:10.3389/fnbeh.2016.00117 (2016).
- 58 Cheng, Y., Lee, S. Y., Chen, H. Y., Wang, P. Y. & Decety, J. Voice and emotion processing in the human neonatal brain. *J Cogn Neurosci* 24, 1411-1419, doi:10.1162/jocn_a_00214 (2012).

17. Line 375 (p.19) – The authors mention that they did not assess aggressive behavior in the participants; a justification of why they did not assess measures of aggression would be warranted, considering the numerous times aggressive behaviour (particular in males) is mentioned in the introduction and discussion.

- Many thanks to the reviewer for the thoughtful suggestion. A justification for the absence of aggression measures was complemented accordingly.
- We also carefully edited the whole manuscript to balance the weight of anxiety and aggression accordingly.
- **Discussion:**
“...Finally, due to our original goal being that of exploring the endophenotypes for anxiety and moral attitudes, we did not assess aggressive behavior in none of the subjects used for this study; thus, a link between the MAOA-uVNTR gene and aggressive behavior...”

REVIEWERS' COMMENTS:

Reviewer #2 (Remarks to the Author):

The authors have addressed all my concerns and suggestions. I recommend the manuscript for publication.

Reviewer #3 (Remarks to the Author):

The authors have adequately addressed my questions and concerns and I have no further comments or suggestions.

The revised paper is much improved from the previous version.